# Visual Abductive Reasoning Meets Driving Hazard Prediction: Problem Formulation and Dataset

## Abstract

This paper addresses the problem of predicting hazards that drivers may encounter while driving a car. We formulate it as a task of anticipating impending accidents using a single input image captured by car dashcams. Unlike existing approaches to driving hazard prediction that rely on computational simulations or anomaly detection from videos, this study focuses on high-level inference from static images. The problem needs predicting and reasoning about future events based on uncertain observations, which falls under visual abductive reasoning. To enable research in this understudied area, a new dataset named the DHPR (Driving Hazard Prediction and Reasoning) dataset is created. The dataset consists of 15K dashcam images of street scenes, and each image is associated with a tuple containing car speed, a hypothesized hazard description, and visual entities present in the scene. These are annotated by human annotators, who identify risky scenes and provide descriptions of potential accidents that could occur a few seconds later. We present several baseline methods and evaluate their performance on our dataset, identifying remaining issues and discussing future directions. This study contributes to the field by introducing a novel problem formulation and dataset, enabling researchers to explore the potential of multi-modal AI for driving hazard prediction.

## 1 Introduction

In this paper, we consider the problem of predicting future hazards that drivers may encounter while driving a car. Specifically, we approach the problem by formulating it as a task of anticipating an impending accident using a single input image of the scene in front of the car. An example input image is shown in Fig. 1, which shows a taxi driving in front of the car on the same lane, and a pedestrian signalling with their hand. From this image, one possible reason is that the pedestrian may be attempting to flag down the taxi, which could then abruptly halt to offer them a ride. In this scenario, our car behind the taxi may not be able to stop in time, resulting in a collision. This simple example shows that predicting hazards sometimes requires abductive and logical reasoning.

Thus, our approach formulates the problem as a visual abductive reasoning [15, 21] from a single image. As an underlying thought, we are interested in leveraging recent advances in multi-modal AI, such as visual language models (VLMs) [1, 19, 43, 9, 22, 25]. Despite the growing interest in self-driving and driver assistance systems, little attention has been paid to the solution we consider here, to the best of our knowledge. Existing approaches rely on predicting accidents through computational simulations using physics-based or machine-learning-based models of the surrounding environment [34]. For instance, they predict the trajectories of pedestrians and other vehicles. Another approach

Submitted to the 37th Conference on Neural Information Processing Systems (NeurIPS 2023) Track on Datasets and Benchmarks. Do not distribute.

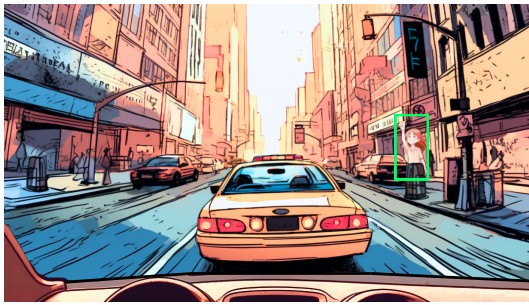

Figure 1: Example of driving hazard prediction from a single dashcam image. The pedestrian in the green box may be attempting to flag down a taxi, and the taxi may abruptly stop in front of our car to offer them a ride.

formulates the problem as detecting anomalies from input videos [36, 37]. However, these methods, which rely only on a low-level understanding of scenes, may have limitations in predicting future events that occur over a relatively long time span, as demonstrated in the example above.

An important note is that our approach uses a single image as input, which may seem less optimal than using a video to predict hazards encountered while driving. There are two reasons simplifying the problem for our choice. First, human drivers are capable of making accurate judgments even from a static scene image, as demonstrated in the example above. Our study is specifically tailored for this particular type of hazards. Humans are apparently good at anticipating the types of hazards that may occur and further estimating the likelihood of each one. Second, there are technical challenges involved in dealing with video inputs. Unlike visual inference from a static image (e.g., visual question answering [2]), there is currently no established approach in computer vision for performing high-level inference from dynamic scene videos; see [21, 15] for the current state-of-the-art. While videos contain more information than single images, we believe that there remains much room to explore in using single-image inputs.

To investigate this understudied approach to driving risk assessment, we present a formulation of the problem and create a dataset for it. Since actual car accidents are infrequent, it is hard to collect a large number of images or videos of real accidents. To cope with this, we utilize existing datasets of accident-free images captured by dashcams, specifically BDD100K (Berkeley DeepDrive) [41] and ECP (EuroCity Persons) [6]; they were originally created for different tasks, e.g., object detection and segmentation. From these datasets, we have human annotators first identify scenes that potentially pose risks, in which an accident could occur a few seconds later. We then ask them to provide descriptions of the hypothesized accidents with mentions of entities (e.g., traffic signs, pedestrians, other cars, etc.) in the scene.

The proposed dataset, named DHPR (Driving Hazard Prediction and Reasoning), is summarized as follows. It contains 15K scene images, for each of which a tuple of a car speed, a description of a hypothesized hazard, and visual entities appearing in the image are provided; see Fig. 2. There are at least one and up to three entities in each scene, each represented by a bounding box with its description. Each entity is referred to as 'Entity #$n$' with $n(=1, 2, 3)$ in the hazard description.

Based on the dataset, we examine the task of inferring driving hazards using traffic scene images. This task involves making inferences based on uncertain observations and falls under the category of visual abductive reasoning, which has been the subject of several existing studies [15, 21, 34]. These studies have also introduced datasets, such as Sherlock [15], VAR [21], and VCR [42]. However, our study differs from these previous works in several aspects, which are outlined in Table 1. While our focus is limited to traffic scenes, our task involves a broader visual reasoning setting that necessitates recognizing multiple objects, understanding their interactions, and engaging in reasoning across multiple steps. Moreover, numerous studies on traffic accident anticipation have generated datasets with similar dashcam imagery, including CCD [3], DoTA [36], A3C [37], and DAD [7]. However,

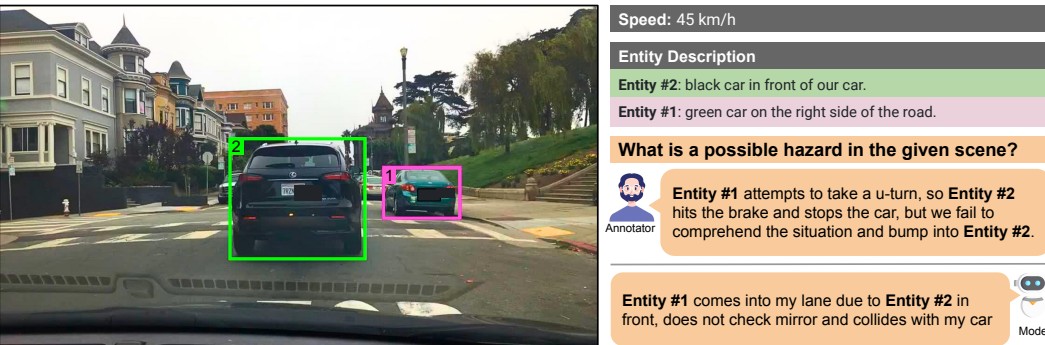

Figure 2: An example from our dataset, DHPR (Driving Hazard Prediction and Reasoning). Each image is annotated with the speed of a car, bounding boxes and descriptions of visual entities involved in a hypothesized hazard, and a natural language explanation of the hazard. The visual entities are referred to as 'Entity #$n$' in the explanation.

Table 1: Comparison of DHPR with existing datasets.

| Dataset | Visual Inputs | Research Problem | Multiple Bboxes | Multi-step reasoning | Object Relationship | Annotation Type |
|---------|---------------|------------------|-----------------|---------------------|---------------------|-----------------|
| Sherlock [15] | Scene images | Abductive reasoning of an interested object | ✗ | ✗ | None | Natural language |
| VAR [21] | Scene images | Abductive reasoning of a missing event | ✗ | ✓ | Event relations | Natural language |
| VCR [34] | Scene images | Commonsense reasoning | ✓ | ✓ | Object interactions | Natural language |
| CCD [3] | Dash-cam videos | Classification of a future event | ✓ | ✗ | Trajectory only | Pre-defined class |
| DoTA [36] | Dash-cam videos | Classification of a future event | ✓ | ✗ | Trajectory only | Pre-defined class |
| **Ours** (**DHPR**) | Dash-cam images | Abductive reasoning of a future event | ✓ | ✓ | Object interactions | Natural language |

these datasets only provide annotations for closed-set classes of accidents/causations. In contrast, our dataset includes annotations for open-set driving hazards expressed in natural language texts.

The following section provides a more detailed discussion of related work (Sec. 2). We then proceed to explain the process of creating the dataset (Sec. 3). Next, we explore various task designs that can be examined using this dataset (Sec. 4). The experimental results, which evaluate the performance of current baseline methods for vision and language tasks in predicting driving hazards, are presented in Sec. 5. Finally, we conclude our study in Sec. 6.

## 2 Related Work

### 2.1 Traffic Accident Anticipation

Traffic accident anticipation has received significant attention in the fields. We focus here exclusively on studies that utilize a dash board camera as the primary input source. The majority of these studies employ video footage as input, which aligns with the task's nature. Most researchers aim to predict the likelihood of an accident occurring within a short time frame based on the input video. It is crucial for the prediction to be both accurate and early, quantified by the time to accident (TTA).

Many existing studies formulate the problem as video anomaly detection. While some studies consider supervised settings [7, 18, 31, 3], the majority consider unsupervised settings, considering the diversity of accidents. Typically, moving objects are first detected in input videos, such as other vehicles, motorbikes, pedestrians, etc., and then their trajectories or future locations are predicted to identify

anomalous events; more recent studies focus on modelling of object interactions [14, 12, 17, 36]. Some studies consider different problem formulations and/or tasks, such as predicting driver's attention in accident scenarios [11], using reinforcement learning to learn accident anticipation and attention [4], and understanding traffic scenes from multi-sensory inputs by the use of heterogeneous graphs representing entities and their relation in the scene [26].

Many datasets have been created for the above research, which contains from 600 to over 4000+ dashcam video recordings, e.g., [13, 7, 18, 3, 37, 36]. However, they provide relatively simple annotation, i.e., if and when an accident occurs in an input video. While some provide annotations for the causes and/or categories of accidents [3, 36, 39], they only consider a closed-set of accident causes and types. On the other hand, the present study considers natural language explanations annotated freely by annotators, leading to encompassing an open set of accident types and causations. It aims to predict potential hazards that may lead to accidents in the near future. The prediction results are not intended to trigger immediate avoidance actions, such as sudden braking, but rather increase the awareness of the risk level and promote caution.

## 2.2 Visual Abductive Reasoning

Abductive reasoning, which involves inferring the most plausible explanation based on partial observations, initially gained attention in the field of NLP [15, 21, 16, 40]. While language models (LMs) are typically adopted for the task, some studies incorporate relative past or future information as context to cope with the limitation of LMs that are conditioned only on past context [28]. Other researchers have explored ways to enhance abductive reasoning by leveraging additional information. For example, extra event knowledge graphs have been utilized [10] for reasoning that requires commonsense or general knowledge, or general knowledge and additional observations are employed to correct invalid abductive reasoning [27]. However, the performance of abductive reasoning using language models exhibits significant underperformance, particularly in spatial categories such as determining the spatial location of agents and objects [5].

Visual abductive reasoning extends the above text-based task to infer a plausible explanation of a scene or events within it based on the scene's image(s). This expansion goes beyond mere visual recognition and enters the realm of the "beyond visual recognition" paradigm. The machine's ability to perform visual abductive reasoning is tested in general visual scenarios. In a recent study, the task involves captioning and inferring the hypothesis that best explains the visual premise, given an incomplete set of sequential visual events [21]. Another study formulates the problem as identifying visual clues in an image to draw the most plausible inference based on knowledge [15]. To handle inferences that go beyond the scene itself, the authors employ CLIP, a multi-modal model pre-trained on a large number of image-caption pairs [30].

## 3 Details of the DHPR (Driving Hazard Prediction and Reasoning) Dataset

### 3.1 Specifications

DHPR provides annotations to 14,975 scene images captured by dashcams inside cars running on city streets, sourced from BDD100K (Berkeley Deepdrive) [41] and ECP (EuroCity Persons) [6]. Each image $x$ is annotated with

- Speed $v$: a hypothesized speed $v(\in \mathbb{R})$ of the car

- Entities $\{e_n = (e_{\mathrm{bbox},n}, e_{\mathrm{desc},n})\}_{n=1,\ldots,N}$: up to three entities ($1 \leq N \leq 3$) leading to a hypothesized hazard, each annotated with a bounding box $e_{\mathrm{bbox},n}$ and a description $e_{\mathrm{desc},n}$ (e.g., 'green car on the right side of the road')

- Hazard explanation $h$: a natural language explanation $h$ of the hypothesized hazard and how it will happen by utilizing the entities $\{e_n\}_{n=1,\ldots,N}$ involved in the hazard; each entity appears in the format of 'Entity #$n$' with index $n$.

Table 2: Split of DHPR. Direct and indirect indicate the type of hypothesized hazards. See text for details.

| Split | Train Set | Validation Set | | Test Set | |
|---|---|---|---|---|---|
| | | Direct | Indirect | Direct | Indirect |
| # | 10,975 | 1,000 | 1,000 | 1,000 | 1,000 |

Table 2 shows the construction of the dataset. In total, there are 14,975 images, which are divided into train/validation/test splits of 10,975/2,000/2,000, respectively.

The validation and test splits are subdivided into two categories based on the nature of the hazards involved. The first category comprises *direct* hazards, which can be predicted *directly*. These hazards are hypothetically caused by a single entity and can be anticipated through a single step of reasoning. The second category includes *indirect* hazards, which require more prediction efforts. These hazards necessitate multiple reasoning steps and are often associated with multiple entities present in the scenes. This classification allows for a comprehensive analysis of models' performance across various aspects. It is important to note that training images do not include direct/indirect tags.

## 3.2 Annotation Process

We employ Amazon Mechanical Turk (MTurk) to collect the aforementioned annotations. To ensure the acquisition of high-quality annotations, we administer an exam resembling the main task to identify competent workers and only qualified individuals are invited to participate in the subsequent annotation process. We employ the following multi-step process to select and annotate images from the two datasets, BDD100K and ECP. Each step is executed independently; generally, different workers perform each step on each image; see the supplementary material for more details.

In the first step, we employ MTurk to select images that will be utilized in the subsequent stages, excluding those that are clearly devoid of any hazards. This leads to the choice of 25,000 images from BDD100K and 29,358 images from ECP. For each image, the workers also select the most plausible car speed from the predefined set [10, 30, 50+] (km/h) that corresponds to the given input image.

In the second step, we engage different workers to assess whether the car could be involved in an accident within a few seconds, assuming the car is traveling at 1.5 times the annotated speed. The rationale behind using 1.5 times the speed is that the original images are acquired in normal driving conditions without any accidents occurring in the future. By increasing the speed, we enhance workers' sensitivity to the risk of accidents, aiming at the generation of natural and plausible hypotheses. We exclude the images deemed safe, thereby reducing the total number of images from 54,358 to 20,791.

In the third step, we ask the workers to annotate each of the remaining images. Specifically, for each image, we ask a worker to hypothesize a hazard, i.e., a potential accident occuring in a near future, in which up to three entities are involved. We ask them to draw a bounding box and its description for each entity. We finally ask them to provide an explanation of the hazard including how it will occur while referring to the specified entities. The hazard explanation must be at least as long as five words and contain all the entities in the format 'Entity #$n$'. Examples are found in Fig. 2.

Finally, we conduct an additional screening to enhance the quality of the annotations. In this step, we enlist the most qualified workers to evaluate the plausibility of the hazard explanations in each data sample. This process reduces the number of samples from 20,791 to 14,975. These are split into train/val/test sets and further direct/indirect hazard types, as shown in Table 2.

# 4 Task Design and Evaluation

## 4.1 Task Definition

We can consider several tasks of different difficulty levels using our dataset. Each sample in our dataset consists of $(x, v, h, \{e_1, \ldots, e_N\})$, where $x$ is an input image, $v$ is the car's speed, $h$ is a hypothesized hazard explanation, and $e_n = (e_{\text{bbox},n}, e_{\text{desc},n})$ are the entities involved in the hazard.

The most natural and ultimate goal is to approach the problem as text generation, where we generate $h$ as natural language text for a given input image $x$. However, this task is particularly challenging due to the difficulty of generating text for visual abductive reasoning. An intermediate step, simpler approach is to treat it as a retrieval problem, since visual abductive reasoning is an emerging field, as demonstrated in a recent study [15] which pioneered visual abductive reasoning and introduced the Sherlock dataset, utilized the same approach. For this task, we have $\{h_i\}_{i=1,\ldots,K}$, which represents a set of candidate hazard explanations $h_i$'s. Our objective is to rank the $h_i$'s for each input image $x$. A higher ranking for the ground truth $h$ of $x$ indicates better prediction. Models generate a score $s = s(x, h)$ for an image-text pair, with the score $s$ indicating their relevance.

We also need to consider how we handle visual entities. There are different options that affect the difficulty of the tasks. The most challenging option is to require models to detect and identify entities by specifying their bounding boxes in the image. A simpler alternative is to select the bounding boxes from a provided set of candidate boxes in the image. An even simpler method assumes that the correct entities are already given as boxes in the input image. Any of these options can be combined with the generation and retrieval tasks.

In our experiments, we focus on retrieval tasks with the easiest setting for visual entities. Specifically, assuming that the bounding boxes of the entities involved in a hypothesized hazard are provided, we consider two retrieval tasks: image-to-text retrieval and text-to-image retrieval. For the former, we rank a list of given texts based on their relevance to an input image, while for the latter, we perform the opposite ranking. Models represent the mapping from three inputs, an image $x$, a hazard explanation $h$, and the involved entities' boxes $\{e_{\text{bbox},1}, \ldots, e_{\text{bbox},N}\}$ as

$$s = s(x, h, \{e_{\text{bbox},1}, \ldots, e_{\text{bbox},N}\}). \tag{1}$$

It is important to note that specifying the bounding boxes of the entities involved helps reduce the inherent ambiguity in hazard prediction. In a given scene, there can be multiple hypotheses of potential hazards. Specifying the entities narrow downs the choices available to the models.

## 4.2 Evaluation Procedure and Metrics

In our retrieval tasks, the models provide a relevance score, denoted as $s$, for an input tuple. We organize our dataset into four splits: val-direct, val-indirect, test-direct, and test-indirect, each containing 1,000 samples, as summarized in Table 2. During evaluation, we treat the direct and indirect types separately. Consider the test-direct split as an example, where we have 1,000 texts and 1,000 images for each hazard type. For image-to-text retrieval, we consider all 1,000 texts that are randomly sampled from all the 2,000 test explanations as candidates and rank them for each of the 1,000 images. Similarly, for text-to-image retrieval, we perform the same ranking process in the opposite direction.

To assess the performance of our models, we employ two metrics. The first metric measures the average rank of the ground-truth (GT) texts for image-to-text retrieval and the average rank of the ground-truth images for text-to-image retrieval. The second metric is the Normalized Discounted Cumulative Gain (NDCG) score [23, 29]. We calculate NDCG scores for the top 200 out of 1,000 hazard explanations. In this calculation, we utilize ChatGPT (gpt-3.5-turbo) from OpenAI to estimate the semantic similarity between each candidate text and its corresponding ground-truth text; see the supplementary material for details. The estimated similarity serves as the relevance score for each candidate text, which allows us to calculate the NDCG score.

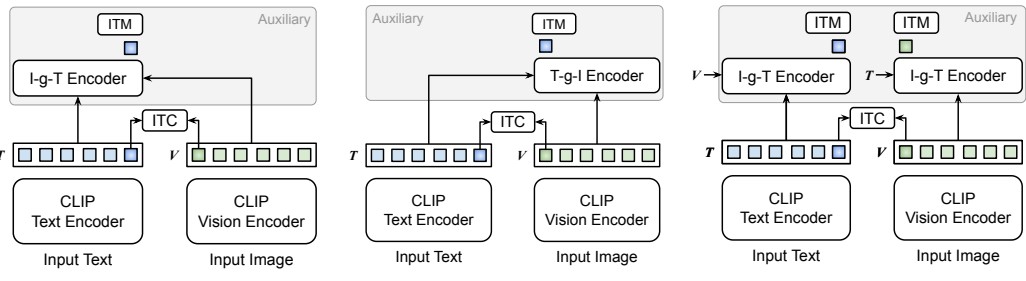

(a) Image-grounded Text Encoder     (b) Text-grounded Image Encoder     (c) Dual-grounded Encoders

Figure 3: The architectures of our CLIP-based baselines with three extensions.

## 5 Experiments

### 5.1 Methods

**How to Input Visual Entities?** To apply a vision and language model to the task under consideration, it needs to calculate a relevance score $s$ for an image $x$ and a hazard explanation $h$ with the bounding boxes of involved visual entities $e_1, \ldots, e_N$. The method is requested to refer to each entity in the hazard explanation in the form of 'Entity #$n$' ($n = 1, 2, 3$). As the entities are specified as bounding boxes in the input image, we need to tell our model which local image regions indicate 'Entity #$n$' ($n = 1, 2, 3$). To do this, we employ an approach to augment the input image $x$ into $\tilde{x}$ with color-coded bounding boxes, following [15, 38]. Specifically, an opaque color is used to represent an image local region under consideration. As there are up to three entities, we employ a simple color-coding scheme, i.e., using purple, green, and yellow colors to indicate Entity #1, 2, and 3, respectively. We employ alpha blending (with 60% opaqueness) between boxes filled with the above colors and the original image; see the supplementary material for more details. We will use $\tilde{x}$ to indicate the augmented image with the specified visual entities in what follows.

**Compared Methods** We experimentally compare several models for vision and language tasks; see Table 3. We adopt CLIP [30] as our baseline method, following the approach in [15]. We employ the model with ViT-B/16 or ViT-L/14 for the visual encoder and BERT-base for the text encoder. In addition, we explore three extended models, which are illustrated in Fig. 3. The first model extends CLIP with an auxiliary image-grounded text encoder (Fig. 3(a)). This encoder updates the text features by attending to the CLIP visual features. The second model utilizes a text-grounded image encoder (Fig. 3(b)). Lastly, the third model combines both text-grounded and image-grounded encoders (Fig. 3(c)). All auxiliary encoders share a simple design, consisting of two standard transformer layers. Each transformer layer includes a self-attention sub-layer and a cross-attention sub-layer, arranged sequentially. Furthermore, we evaluate two popular existing methods for vision and language tasks: UNITER [8] and BLIP [20]. UNITER employs a single unified transformer that learns joint image-text embeddings. It uses a pre-trained Faster R-CNN to extract visual features. BLIP employs two separate transformers, namely a Vision Transformer for visual embeddings and a BERT Transformer for text embeddings. For all the models but UNITER, we employ the cosine similarity between the image and text embeddings as the relevance score; UNITER has a retrieval head to yield a score.

### 5.2 Training

**Loss Functions** To train (or fine-tune) the above models, we employ two training objectives (i.e., loss functions). One is the contrastive loss over a set of image-text pairs [30] and the other is the matching loss between an image and a text [24], if applicable. See the supplementary material for details.

Table 3: Comparison of average ranks of GT texts and NDCG scores (in brackets if applicable) on the test split. Lower ranks indicate better performance, while higher NDCG scores indicate better.

| Model | Visual Encoder | Text-to-Image | | Image-to-Text | |
|---|---|---|---|---|---|
| | | Direct | Indirect | Direct | Indirect |
| Random | - | 500 | 500 | 500 | 500 |
| UNITER [8] | Faster R-CNN | 172.3 | 186.5 | 173.8 (74.2) | 181.2 (71.9) |
| BLIP [20] | ViT-B/16 | 153.4 | 172.1 | 151.9 (78.6) | 176.1 (72.3) |
| BLIP2 [19] | ViT-L/14 | 98.9 | 82.5 | 94.3 (74.9) | 81.1 (71.6) |
| Baseline | ViT-B/16 | 77.2 | 75.3 | 78.4 (81.8) | 73.3 (79.2) |
| w/ Text Encoder | ViT-B/16 | 75.9 | 73.5 | 73.2 (82.2) | 68.1 (80.3) |
| w/ Image Encoder | ViT-B/16 | 74.5 | 72.2 | 79.1 (81.4) | 69.7 (80.3) |
| w/ Dual Encoders | ViT-B/16 | 74.8 | 70.2 | 69.2 (82.9) | 64.3 (80.4) |
| w/ Dual Encoders | ViT-L/14 | **65.9** | **55.8** | **66.5 (84.4)** | **53.8 (80.7)** |

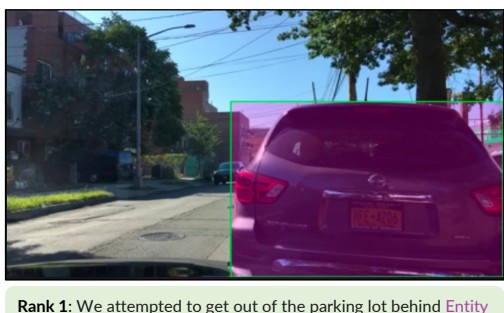
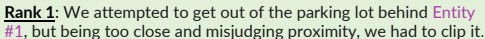
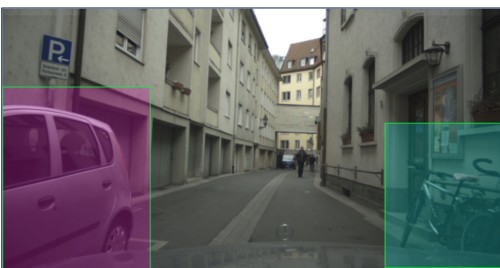

GT — **Rank 1**: We attempted to get out of the parking lot behind Entity #1, but being too close and misjudging proximity, we had to clip it.

GT — **Rank 5**: I adjust car to my right, so I don't hit Entity #1 but instead my car hits Entity #2 due to narrow pass

top — **Rank 2 (0.4)**: Entity #1 is parked very close to me; even though I'm moving my car slowly, getting out of my parking spot will scratch Entity #1
**Rank 3 (0.6)**: our car is very near to Entity #1 at the given speed we could hit on the back of Entity #1 within a second.
**Rank 4 (0.2)**: Entity #1 is very close to me, so even though I am driving slowly due to traffic, I will hit Entity #1

top — **Rank 1 (0.4)**: Entity #1 and Entity #2 makes the road narrow for my car, due to this, I pull to the left more and hits Entity #1
**Rank 2 (1.0)**: Entity #1 and my car converge at the same time at Entity #2, due to this my car clips Entity #1 due to Entity #2
**Rank 3 (1.0)**: Entity #1 is coming in my direction, and due to Entity #2, my car clips Entity #1 on the side.

(a) An example of **direct hazard** prediction

(b) An example of **indirect hazard** prediction

Figure 4: Examples of the image-to-text retrieval by the best-performing baseline model, including the annotated hazard (GT) and its rank, alongside the other top three candidates. Each candidate's rank is indicated as **Rank n** with the brackets containing its ChatGPT similarity to the GT.

**Entity Shuffle Augmentation** While a hypothesized hazard explanation can contain multiple visual entities, their order in the explanation is arbitrary, e.g., 'Entity #1' may appear after 'Entity #2' etc in the text. As explained earlier, we assign a color to each index ($n = 1, 2, 3$), and this assignment is fixed throughout the experiments, i.e., purple = 'Entity #1,' green = 'Entity #2,' and yellow= 'Entity #3.' To facilitate the models to learn this color coding scheme, we augment each training sample by randomly shuffling the indices of entities that appear in the explanation, while we keep the color coding unchanged.

## 5.3 Results and Discussions

Table 3 presents the results of the compared methods for the retrieval tasks. Several observations can be made. Firstly, regardless of the retrieval mode (text-to-image or image-to-text), the performance is generally better for indirect hazard types compared to direct ones. This difference in performance can be attributed to the nature of the hazard types. Direct hazards are simpler and have annotations that are more similar to each other, whereas indirect hazards are more complex, leading to more diverse and distinctive annotations. Secondly, the ranks of the GT (ground-truth) texts are well aligned with the NDCG score, indicated within parentheses for image-to-text retrieval.

Thirdly, our baseline models, which are based on CLIP, demonstrate superior performance (i.e., average GT rank ranging from 53.8 to 79.1) compared to UNITER, BLIP and BLIP2 (i.e., ranging from 81.1 to 186.5). This may be attributable to the larger-scale training of CLIP using diverse image-caption pairs. Additionally, we observe that the best performance is achieved by the model that utilizes dual auxiliary encoders and a larger ViT-L/14 vision encoder. This finding suggests that the task at hand is highly complex, requiring models with sufficient capacity to handle this complexity. In summary, our results indicate that it is possible to develop better models for this task.

It is important to note that even the best-performing model achieves an average rank of around 60 out of 1,000 candidates, which may not appear impressive. However, average ranks may not accurately represent the true performance of models, although they are effective for comparing different models. This is because different scene images can have similar hazard hypotheses and explanations, as shown in Fig. 4, due to the nature of driving hazards. Additionally, the same scenes can have multiple different hazard hypotheses due to the nature of abductive reasoning. While our experiments limit the number of hypotheses by specifying participating visual entities, it may not reduce the possible hypotheses to just one. These observations imply that the top-ranked hazard explanations by a model can still be practically useful, even if they result in seemingly suboptimal ranking scores. Therefore, it may be more appropriate to use the NDCG score as the primary metric to assess the real-world performance of models.

## 6   Conclusion and Discussions

We have introduced a new approach to predicting driving hazards that utilizes recent advancements in multi-modal AI, to enhance methodologies for driver assistance and autonomous driving. Our focus is on predicting and reasoning about driving hazards using scene images captured by dashcams. We formulate this as a task of visual abductive reasoning.

To assess the feasibility and effectiveness of our approach, we curated a new dataset called DHPR (Driving Hazard Prediction and Reasoning). This dataset comprises approximately 15,000 scene images captured by dashcams, sourced from existing datasets initially designed for different tasks. To annotate each scene image, we employed a crowdsourcing platform. The annotations include the car's speed, a textual explanation of the hypothesized hazard, and visual entities involved in the hazard, represented by bounding boxes in the image along with corresponding descriptions in text format.

Next, we designed specific tasks utilizing the dataset and introduced proper evaluation metrics. we conducted experiments to evaluate the performance of various models, including a CLIP-based baseline and popular vision and language (V&L) models, on image-to-text and text-to-image retrieval tasks in the setting that participating visual entities are assumed to be given. The experimental results demonstrate the feasibility and effectiveness of the proposed approach while providing valuable insights for further investigations.

It should be emphasized that while there are numerous studies on predicting traffic accidents, our approach tackles a different problem. Previous research primarily aims to directly forecast the occurrence of accidents, with the objective of prevention. In contrast, our study is geared towards predicting potential hazards that could eventually lead to accidents in the future. While the outcomes of our prediction may not necessitate immediate avoidance actions, such as abrupt braking, they serve to make drivers aware of the magnitude of the risk and encourage them to pay attention. This will be useful for driver assistance systems.

This area remains largely unexplored within the related fields, offering numerous opportunities for further research. One promising direction is the application of LLMs to the problem. LLMs are now recognized for their ability in hypothesis generation, multi-step reasoning, and planning [33, 32, 35]. Leveraging these capabilities, along with their extension to multi-modal models [1, 19, 43, 22] holds great potential. As this unfolds, our dataset will continue to be relevant for studying the creation of reasoning texts.

Another direction for future exploration involves expanding the study from static images to videos. While static images provide sufficient information for predicting and reasoning about a wide but limited range of hazards, incorporating temporal information from videos could provide additional clues, enabling the consideration of a broader range of hazards and potential accidents. Without our intermediate step of leveraging a single image-based method, it would be difficult to navigate the complexities of video-based prediction.

In conclusion, we have high hopes that our study and dataset will spark the interest of researchers and contribute to the advancement of driver assistance and autonomous driving systems.

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
