# Supplementary Material for "Visual Abductive Reasoning Meets Driving Hazard Prediction: Problem Formulation and Dataset"

## A    Access to the DHPR (Driving Hazard Prediction and Reasoning) Dataset

### A.1    URLs

The DHPR dataset can be downloaded at

- https://github.com/DHPR-dataset/DHPR-dataset

It contains annotations paired with images sourced from two external datasets, BDD100K [7] and ECP [1]. Users must download the images from their respective sources:

- https://bdd-data.berkeley.edu/
- https://eurocity-dataset.tudelft.nl/eval/overview/statistics

We have created a website that allows reviewers to browse through the dataset:

- https://huggingface.co/spaces/DHPR/Demo

### A.2    Notes on Availability and Maintenance of the Data

The DHPR dataset created in this study is openly available for access from the URL given above. The dataset is provided in both training and validation sets. It is our commitment to continually update and maintain the availability of the dataset. Additionally, we plan to establish an evaluation server and leaderboard in the future. Any updates pertaining to the dataset will be communicated through the aforementioned repository, ensuring that users have access to the most up-to-date information.

### A.3    Ethical and Responsible Use

The present study complies with the ethical standards for responsible research practice. Our dataset is built upon images of two existing datasets, ECP and BDD100K. It is compliant with GDPR for ECP [1] and other data-related regulations for BDD100K [7]. We protected the anonymity of personal information by blurring identifiable details in the images used in both the main paper and this supplementary material. The datasets are sourced following the licensing regime of each dataset.

Submitted to the 37th Conference on Neural Information Processing Systems (NeurIPS 2023) Track on Datasets and Benchmarks. Do not distribute.

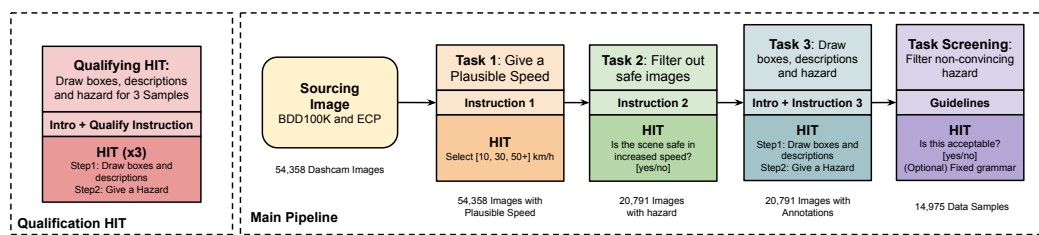

Figure 1: Overview of the data collection process

## B  Data Collection Process

### B.1  Overview

We used Amazon Mechanical Turk to generate the dataset. Figure 1 provides an overview of the data collection process. The process involves two steps: the preliminary step for qualifying workers and the main pipeline for creating annotations. Only the workers filtered in this qualification step were invited to participate in the main pipeline (Sec. B.2).

The main pipeline consists of three tasks:

- **Task 1**: The car's speed is estimated for each input scene image (Sec. B.3);

- **Task 2**: The possibility of an accident occuring is estimated so that risk-free scene images are removed (Sec. B.4);

- **Task 3**: A driving hazard is hypothesized and annotated for an input image (Sec. B.5).

We employ these three steps for the following reasons. In Task 3, workers are asked to annotate a hypothesized hazard for each input image. Since the original images are of accident-free scenes, it can often be challenging to identify hazards, even as hypotheses. If workers are given the freedom to choose whether or not to annotate an image based on its difficulty, we will struggle to collect a sufficient amount of annotations. However, it is also problematic to force workers to identify hazards in risk-free scenes. Therefore, we designed Task 2 to address this issue. In Task 2, we remove scene images with minimal risk and transfer the remaining images to Task 3. To avoid removing too many images, we ask workers to assess the risk of the scene images by assuming a 50% increase in the car's speed. We designed Task 1 to estimate the normal speed of the car.

For each task in the qualification step and the main pipeline, we created HITs (Human Intelligence Tasks). All communication was conducted in English. The dataset was collected and evaluated from January through March 2023. Throughout the process, we paid the workers around $10 USD/hour. To be specific, we paid $0.02 USD per HIT for multiple choice HITs such as Task 1 and Task 2. For Task 3, we paid $0.2 USD per HIT, and some workers may finish this within 40 seconds.

We input 54,358 images from two datasets, BDD100K and ECP, to the main pipeline, followed by an additional qualification step (Sec. B.2). Consequently, we acquired the annotations for a total of 14,975 images, which comprise the final DHPR dataset.

### B.2  Preliminary Step: Qualification/Screening of Workers

As mentioned above, we utilized a qualifying test to identify competent workers. This test not only serves as an evaluation tool but also provides potential workers with an overview of the tasks discussed earlier. On the initial page, we present essential information in the form of clickable/expandable items within a menu. This includes a description of the qualifying task (Fig.2), instructions on annotating visual entities (Fig.3), guidelines for writing effective hazard explanations (Fig.4), and examples of exemplary annotations (Fig.5).

The qualifying task is designed to mimic Task 3. Its purpose serves three objectives: (i) to assess the workers' understanding of the instructions, (ii) to evaluate their experience and comprehension of driving cars and traffic conditions, and (iii) to gauge their proficiency in providing annotations in the

**Instructions for the qualifying Task (click to expand/collapse)**

Thank you for participating in this HIT!

Please read the following rules carefully and check the **examples** as the references

Answers that do not comply with the rules **will be rejected**.

Also, answers using Automation tools will be rejected for all of the tasks

**Your task:**

In this task, we would like you to make an inference about a traffic accident that would occur in a few seconds. You are given an image and your car's speed, which will be used to draw boxes and make an explanation.

Specifically, we ask you to **draw a bounding box for each entity** that would be involved in an accident, **write object descriptions**, and **write down a rationale for the accident involving your car**.

**Note** that each image is selected and identified as a potential accident image by human annotators. It means that there would be the possibility of a traffic accident in a few seconds. It is mandatory to write down a most probable accident rationale.

**You will do these in two steps**.

**Step 1: Draw a bounding box for each entity that would cause a traffic accident and write down an object description**

- Draw a bounding box of the **entity**
- Write down a **description** of that entity
- **Repeat** for all objects

**Step 2: Write down your car's accident rationale involving all the entities.**

- Must use the **"(Entity #) word"** instead of object noun in the accident rationale's input form.
- Imagine yourself **as the driver** driving at a given car speed (i.e., first-person view).
- Considering the given car speed and the surrounding situation, write down an explanation or rationale for the traffic accident that would occur involving your car.

**Rules:**

**For All Steps:**

- Since this is a visual abductive reasoning task (i.e., the process of making the most plausible inference in the face of incomplete information), the inferences do not have to be perfect.
- Please take the given car speed into the account.
- The inference must be related to the selected bounding box.
- Please do not use **the template or pattern language** in the rationale.
- Please make the inference from two perspectives. One is whether your car would directly crash into any cars, pedestrians, or cyclists. The other is whether your car would hit a pedestrian hidden by an object (e.g., a truck or bus); see the first example below.

**For Step 1: Drawing bounding boxes and descriptions**

- Please draw a bounding box on **any entities** (a car, a traffic sign, pedestrians, a part of the road surface, etc.) that would cause the accident.

**For Step 2: Write down your car's accident rationale**

- The rationale is **not an image description**. It is to describe how the entities cause the accident.
- Write a **complete sentence** that has more than five words at least.

**Reasoning based on your actual driving experiences is welcome.**
**Please see the following examples**

Figure 2: Instructions for the qualifying test, which also serves as an introduction to the real tasks in the main annotation pipeline.

required format. Each worker was administered three questions as part of this task. Examples are presented in Fig. 6.

To ensure a diverse range of annotations, we invited over 500 workers worldwide to participate in the qualification test. In order to maintain quality control, we specifically targeted workers with a proven track record of approving more than 10,000 HITs and maintaining an approval rate of over 95%. Following the evaluation of their performance on the test, we manually selected 60 workers who met our criteria. These selected workers were then invited to participate in the main annotation pipeline.

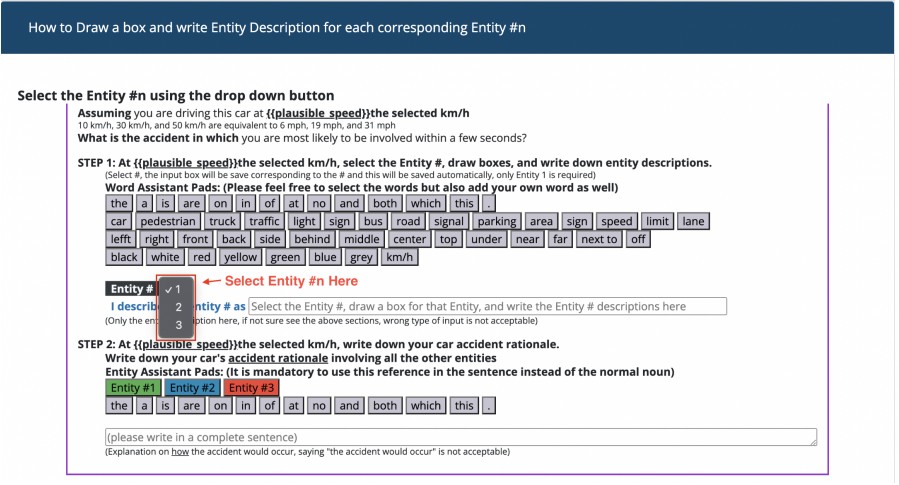

(a) Step 1: Select an entity index

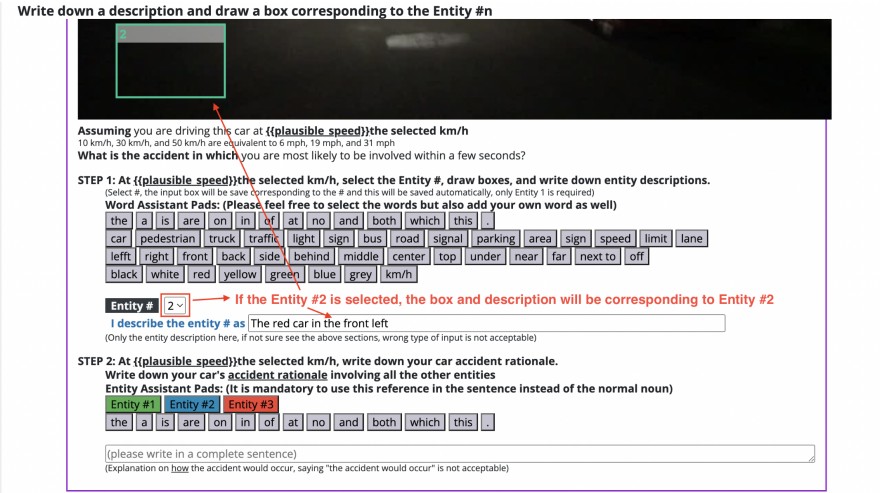

(b) Step 2: Draw a box and input a brief description

Figure 3: Explanation of how to annotate visual entities by using the tools provided.

## B.3 Task 1: Estimating the Car's Speed

The workers were instructed to estimate the speed of the car based on the dashcam image of a scene. The web interface for the HIT of the task is shown in Fig. 7. A total of 54,358 scene images were used, with each assigned to a single worker. As a result of this task, we obtained annotations for all 54,358 images.

## B.4 Task 2: Predicting Accident Possibility to Filter Images

The second task aims to assess the probability of a car being involved in an accident within a few seconds. In order to make Task 3, annotating hypothesized hazards, efficient, it was necessary to eliminate scene images with a very low likelihood of accidents. To achieve this, we introduced an increased speed that was 1.5 times faster than the annotated speed used in the first task. The intention behind this speed increase was to instill a stronger sense of the potential for an accident among the workers, given that the original speed was determined based on their perception of a safe speed. Figure 8 illustrates the instruction and annotation form provided for this task. Consequently, out of

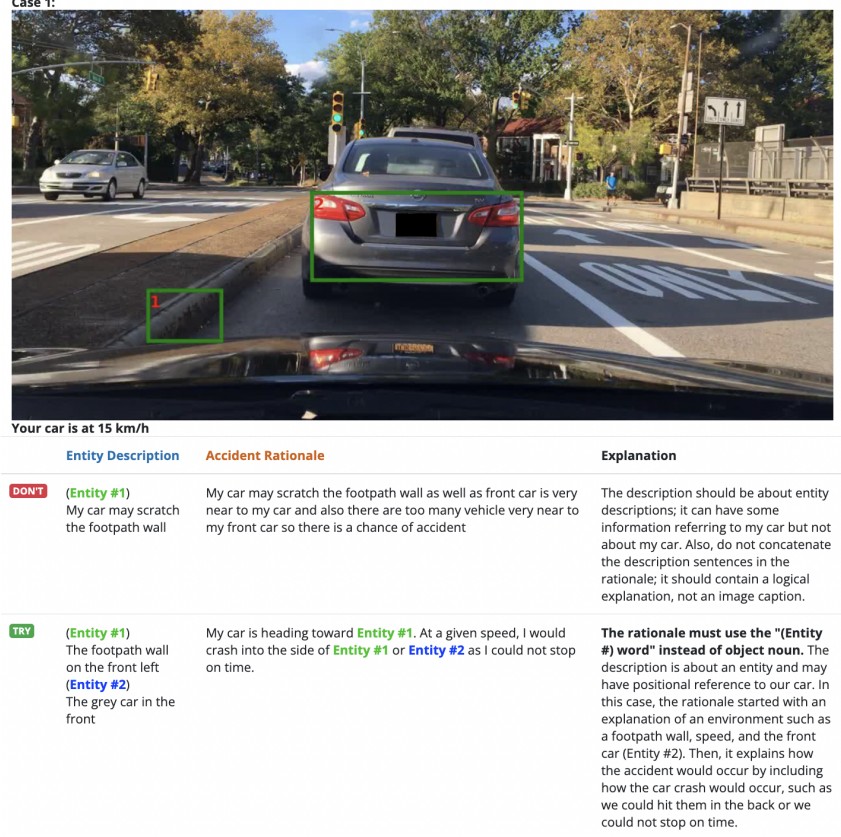

Figure 4: Guidelines for writing effective hazard explanations

the initial set of 54,358 images, 20,791 images were filtered and subsequently used in the following task.

## B.5 Task 3: Hypothesizing and Annotating a Hazard

The final task involves hypothesizing and annotating a hazard for each of the previously filtered images. This task comprises two parts. The first part involves annotating the visual entities associated with the hazard. Workers are instructed to draw a bounding box around each entity and provide a brief description of it. The second part requires providing a natural language explanation of the hazard, using the term 'Entity #$n$' to refer to the involved entities. Figure 9 shows the instruction and annotation forms for this task.

Task 3 is the most time-consuming, accounting for 80% of the total annotation time. Based on our statistics, each worker may spend up to three minutes per image and can complete a maximum of three hundred HITs per day. In order to enhance productivity and reduce inconsistencies in answers, we have implemented data input validation and a user interface assistant[1].

---

[1]Our data input validation system ensures that submissions meet the following criteria: at least one box must be drawn; each box should have a corresponding entity description, and vice versa; only one bounding box per entity is allowed; When adding a box, a new entity must be utilized; and the hazard explanation must be at least five words long. If any of these criteria are not met in a submission, a warning prompt will be displayed.

In addition, the "Word Assistant Pads" feature was provided to the workers to minimize the need for typing, as shown in Fig. 9(b) and (c). It automatically fills in the text prompt input form by clicking buttons. This aid also serves as a reminder to workers regarding the expected content of the input form. Additionally, a brief guideline emphasizing the necessary components of the sentence was provided, including an accident-related entity, its relative position, and the resulting accident. Also, in close proximity to the hazard input form, there

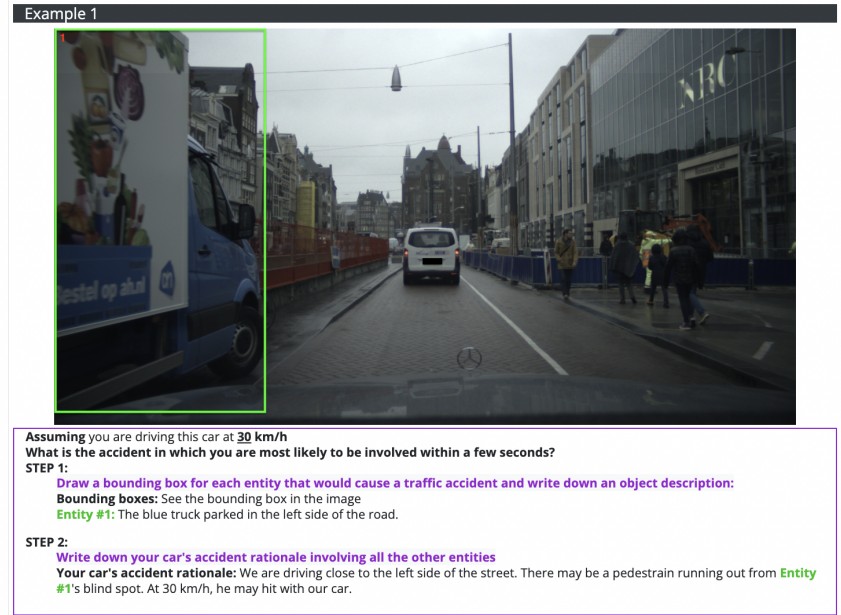

Figure 5: Examples of exemplary annotations

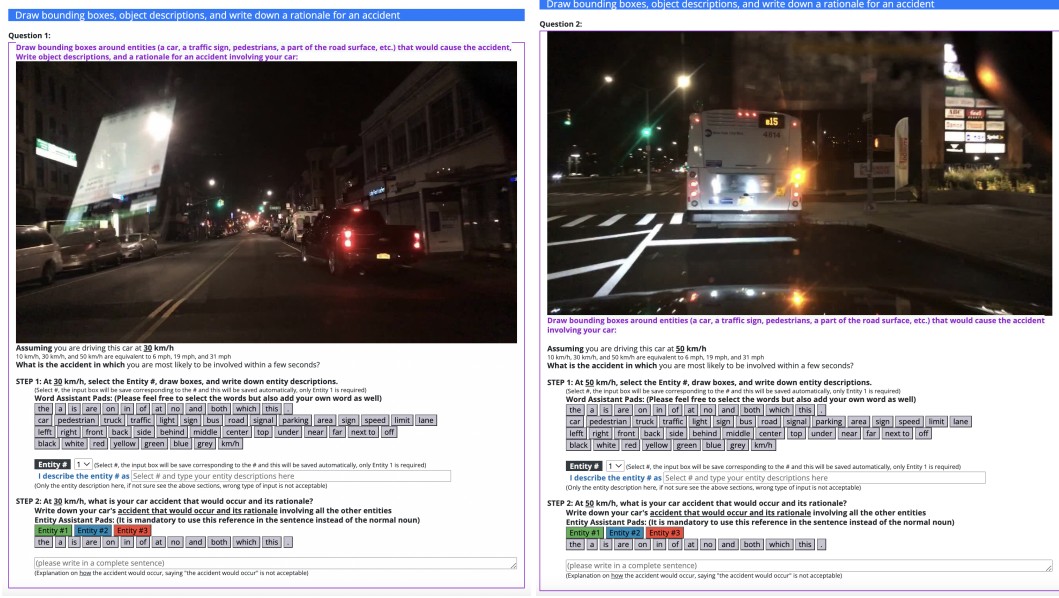

Figure 6: Examples of the qualfifying test

## B.6 Post Process: Data Validation

We checked the results during and after Task 3 ourselves, and removed annotations with obvious errors. Additionally, we invited a small number of the most reliable workers to an additional task of eliminating Task 3 annotations that had unsatisfactory quality. The web page design for the HIT is shown in Fig. 10. The workers were presented with annotations for each scene image, including bounding boxes, descriptions of visual entities, and hazard explanations. They were asked a binary (yes/no) question regarding the acceptability of the annotations. If necessary, the workers were also

___

are reminders for the specific entities ('Entity #1', 'Entity #2', 'Entity #3'), as well as preposition words, to discourage the input of noun words for the entity.

Thanks for participating in this HIT!

Please check **the examples** as it will make your life much easier

Please read the rules carefully.

Answers that do not comply with the rules **will be rejected**.

Using **Automation** will likely be rejected in all of the worker tasks.

**Your task:**

In this task, we would like you to reason **a plausible speed of your car in a given scene**.

**Make inferences about a plausible car speed**

1. Assume that a given image is taken by your car (i.e., first-person view).
2. Reason a plausible (natural) speed of your car by taking into account the information about a situation of a given scene and commonsense.
3. Select the plausible speed of your car from [10 km/h, 30 km/h, 50+ km/h] which is equivalent to [6 mph, 19 mph, 31+ mph]

**Rules:**

- Please select a **reasonable** plausible car speed (i.e., a slow car speed for a crowded area and a high speed in a highway area)

(a) Instructions

**The plausible car speed is** ○10 ○30 ○50+ **km/h**
(The speed choices are equivalent to 6 mph, 19 mph, and 31+ mph)

(b) Annotation form

Figure 7: (a) Instruction and (b) the anotation form for Task 1, which requests the workers to estimate the car's speed.

requested to correct minor mistakes such as grammatical errors or incorrect word choices. Following this screening step, we obtained annotations for a total of 14,975 scene images.

Thanks for participating in this HIT!

Please read the rules carefully.

Answers that do not comply with the rules **will be rejected**.

Please check **the examples** for the references.

Using Automation will likely be rejected in all of the worker tasks.

### Your task:

In this task, we would like you to reason about the **possibility of a traffic accident that would occur within a few seconds** of a given scene.

**Make inferences about the possibility of a traffic accident**

1. Imagine yourself as the driver (i.e., first-person view) driving at a given plausible speed
2. Rate how likely a traffic accident would occur in a few seconds
   - **Yes**: It's a guess, but a traffic accident might occur in a few seconds.
   - **No**: It's obvious that any traffic accidents will not occur.

### Rules:

- Please consider the possibility of accident from two perspectives. One is whether your car would directly crash into any cars, pedestrians, or cyclists. The other is whether your car would hit a pedestrian **hidden by the object** (e.g., truck or bus); see the first example below.

**Please see the following examples**

(a) Instruction

**The plausible car speed is ${plausible_speed} km/h**
10 km/h, 30 km/h, and 50+ km/h are equivalent to 6 mph, 19 mph, and 31 mph
**A traffic accident might occur in a few seconds:** ○No  ○Yes

(b) Annotation form

Figure 8: (a) Instruction and (b) annotation form for Task 2, which is to predict the possibility of an accident for an input image.

Thank you for participating in this HIT!

Please read the following rules carefully and check the **examples** as the references

Answers that do not comply with the rules **will be rejected**.

Also, answers using Automation tools will be rejected for all of the tasks

**Your task:**

In this task, we would like you to make an inference about a traffic accident that would occur in a few seconds. You are given an image and your car's speed, which will be used to draw boxes and make an explanation.

Specifically, we ask you to **draw a bounding box for each entity** that would be involved in an accident, **write object descriptions**, and **write down a rationale for the accident involving your car**.

**Note** that each image is selected and identified as a potential accident image by human annotators. It means that there would be the possibility of a traffic accident in a few seconds.
**You will do these in two steps**.

**Step 1: Draw a bounding box for each entity that would cause a traffic accident and write down an object description**

- Draw a bounding box of the **entity**
- Write down a **description** of that entity
- **Repeat** for all objects

**Step 2: Write down your car's accident rationale involving all the entities.**

- Must use the **"(Entity #) word"** instead of object noun in the accident rationale's input form.
- Imagine yourself as the driver driving at a given car speed (i.e., first-person view).
- Considering the given car speed and the surrounding situation, write down an explanation or rationale for the traffic accident that would occur involving your car.

(a) Instruction

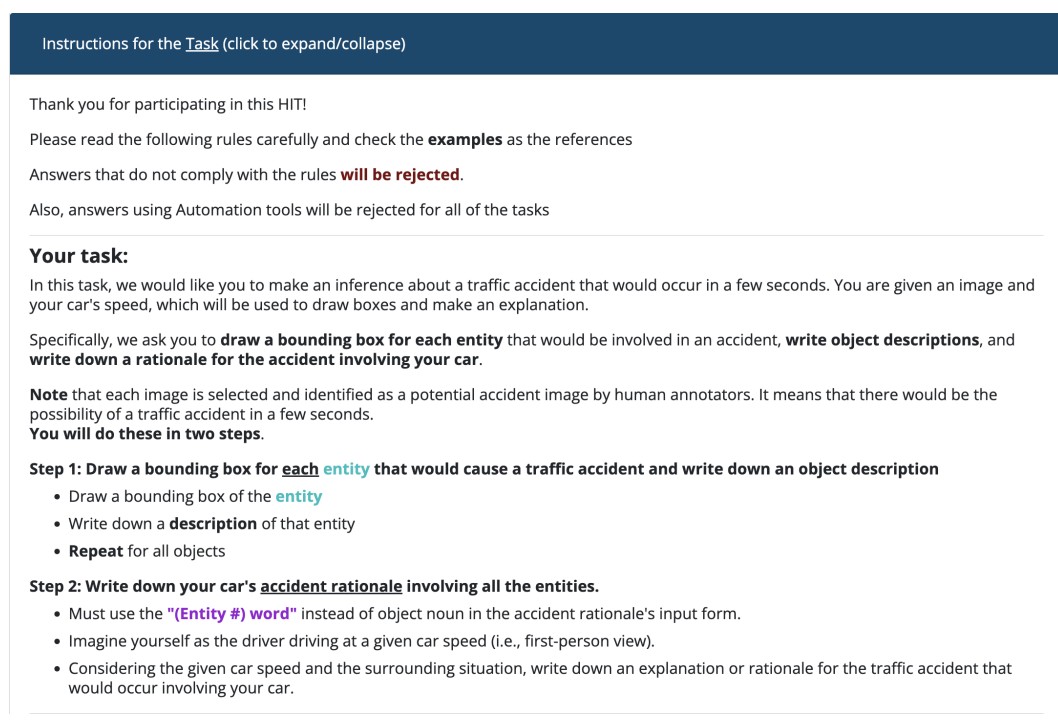

(b) Annotation form for visual entities

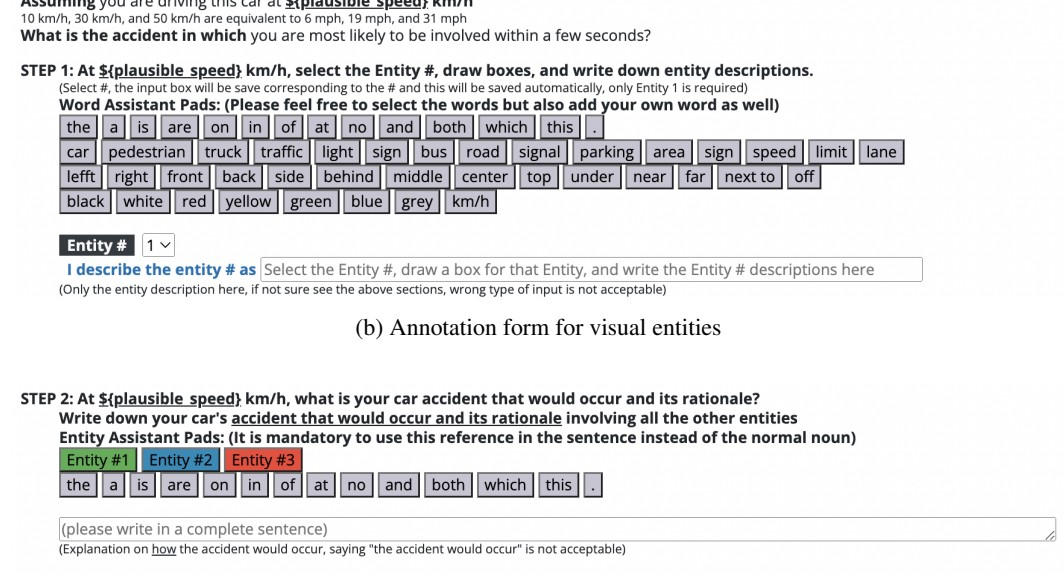

(c) Annotation form for hazard explanations

Figure 9: (a) Instruction, (b) & (c) annotation forms for Task 3, which is to annotate visual entities involved in a hypothesized hazard and provide an explanation of the hazard.

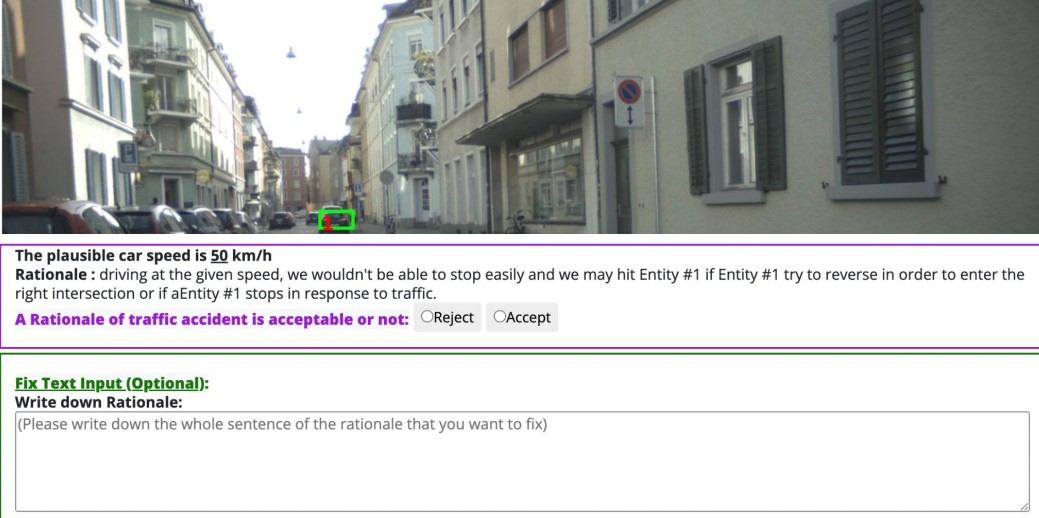

**Check if it is a rationale, and if it makes sense all not**

**Considering:**
- Is this a rationale sentence?, I want the sentence that explain how the accident happen?
- Not a warning sentence
- "**No** If cause"
- The accident is convincing based on your opinion, considering distance, speed and other conditions.

**The plausible car speed is 50 km/h**
**Rationale :** driving at the given speed, we wouldn't be able to stop easily and we may hit Entity #1 if Entity #1 try to reverse in order to enter the right intersection or if aEntity #1 stops in response to traffic.
**A Rationale of traffic accident is acceptable or not:** ○Reject ○Accept

**Fix Text Input (Optional):**
**Write down Rationale:**
(Please write down the whole sentence of the rationale that you want to fix)

Figure 10: Annotation form for the data validation task

## C   Dataset Analysis

The DHPR dataset showcases a diverse range of annotated hazards, which is a result of the various traffic scenes captured in the BDD100K and ECP image sources. It is an initial effort to comprehensively categorize driving hazards based on a rich blend of image sources which may require further refinement in terms of diversity and real-world applicability. Table 1 displays the statistics of word usage in hazard explanations for each data split. It is important to note that our hazard explanations are generally longer compared to those in Sherlock [3], a dataset that focuses on visual abductive reasoning in a broader domain. However, since our explanations specifically relate to driving hazards, they tend to exhibit more similarity to each other when compared to Sherlock. To address this, as explained in the main paper, we have introduced the NDCG score based on ChatGPT. The purpose of this score is to provide a more precise measure of similarity between two explanations. For more details, refer to Sec. G.

Table 1: Statistics of word usage in the hazard explanations

| Split | Avg. Length Tokens | Unique Words | Less than 5 Occurrence Words (%) | More than 10 Occurrence Words (%) |
|---|---|---|---|---|
| Train | 30.9 | 2,341 | 61.9 | 30.5 |
| Val-direct | 26.9 | 851 | 66.3 | 24.0 |
| Val-indirect | 27.6 | 891 | 64.9 | 23.7 |
| Test-direct | 27.0 | 842 | 66.3 | 23.6 |
| Test-indirect | 27.7 | 913 | 65.8 | 23.8 |

Table 2 presents the types of entities that the self-car is described as hitting in the hazard explanations for each data split. The direct type of hazards mainly involve cars as the entities involved, while the indirect type encompasses a broader range of entities. In order to visualize the distribution of verbs used in the explanations for each hazard type, we have included cloud plots in Figs. 11 and 12.

Table 3 provides a detailed statistical breakdown of the dataset. It outlines the division of samples between the training, validation, and test sets while also specifying the source of each image, either from the ECP or BDD dataset. The table categorizes the types of hazards as 'Direct' or 'Indirect' and offers further granularity by showing the speed of our car, the average length of hazard descriptions, and the common position and orientation words used. Additionally, it enumerates the types of entities involved, such as cars and pedestrians, to offer a comprehensive overview of the dataset's composition.

Table 2: Types of entities that the self-car is described as hitting in the hazard explanations.

| Split | Car | Motorbike | Pedestrian | Others |
|---|---|---|---|---|
| val-direct | 901 | 31 | 42 | 26 |
| val-indirect | 762 | 66 | 114 | 58 |
| test-direct | 915 | 26 | 36 | 23 |
| test-indirect | 769 | 70 | 124 | 37 |

Table 3: Detailed Dataset Statistics

| Data Attributes | Training Set | Validation Set | | Test Set | |
|---|---|---|---|---|---|
| | | Direct | Indirect | Direct | Indirect |
| **Data Sample** | 10975 | 1000 | 1000 | 1000 | 1000 |
| **Image sourced from:** | | | | | |
| ECP dataset | 6010 | 456 | 502 | 480 | 472 |
| BDD dataset | 4965 | 544 | 498 | 520 | 528 |
| **Our car speed:** | | | | | |
| 15 km/h | 4467 | 518 | 275 | 533 | 267 |
| 45 km/h | 3859 | 351 | 439 | 338 | 430 |
| 75+ km/h | 2649 | 131 | 286 | 109 | 303 |
| **Hazard:** | | | | | |
| Avg. Length Tokens | 30.9 | 26.9 | 27.6 | 27.0 | 27.7 |
| **Position words:** | | | | | |
| Left | 1347 | 67 | 209 | 70 | 193 |
| Right | 1511 | 59 | 168 | 66 | 168 |
| Front | 1749 | 156 | 102 | 130 | 79 |
| Side | 1563 | 73 | 162 | 80 | 157 |
| Back | 934 | 158 | 65 | 149 | 71 |
| **Accident-related words:** | | | | | |
| Hit | 6608 | 558 | 577 | 602 | 578 |
| Crash | 1382 | 118 | 142 | 94 | 110 |
| Collide | 643 | 56 | 61 | 52 | 63 |
| Clip | 121 | 25 | 75 | 19 | 73 |
| **Entity Descriptions:** | | | | | |
| **Entities:** | | | | | |
| Car | 7427 | 822 | 613 | 848 | 630 |
| Pedestrain and Bicycles | 2233 | 98 | 216 | 75 | 228 |
| **Orientation words:** | | | | | |
| Front | 3921 | 535 | 210 | 509 | 233 |
| Side | 2448 | 87 | 390 | 81 | 436 |
| Right | 2275 | 78 | 408 | 70 | 438 |
| Left | 1719 | 68 | 299 | 63 | 310 |
| Ahead | 1552 | 224 | 184 | 210 | 167 |
| Road | 1380 | 62 | 106 | 63 | 105 |
| Brake | 1223 | 142 | 0 | 141 | 0 |
| Parked | 1027 | 0 | 156 | 0 | 157 |

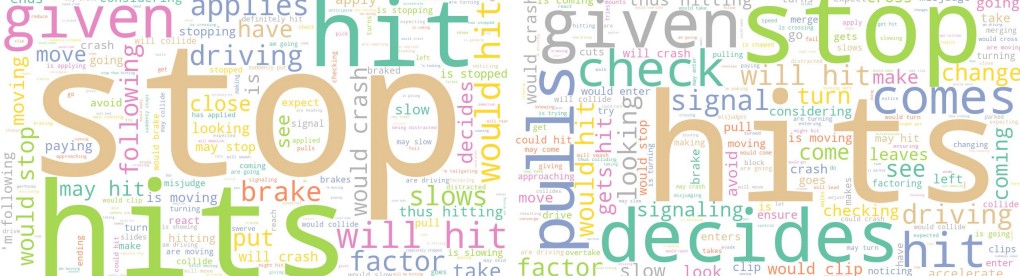

Figure 11: Phrase of Direct Set          Figure 12: Phrase of Indirect Set

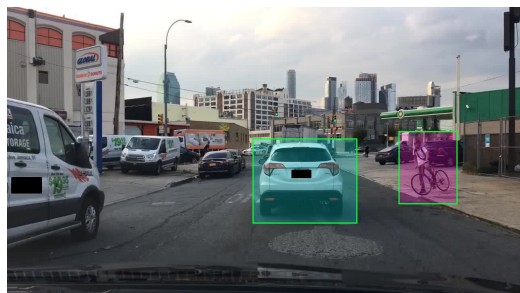 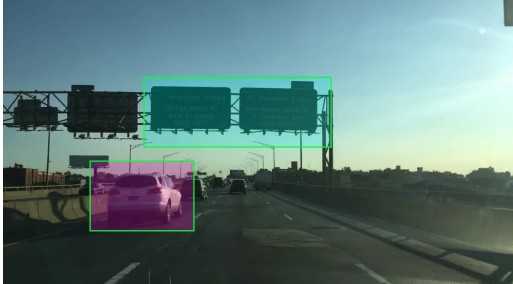

**GT at Rank 1**: Entity #1 decides to go behind of Entity #2 to cross street, misjudges my speed, won't be able to stop in time and hits Entity #1

**Rank 2 (0.7)**: not paying attention due to Entity #2, don't see Entity #1 ahead, i brake late and hits Entity #1 due to my given speed

**Rank 3 (0.8)**: Entity #1 brakes, can't swerve to my left due to Entity #2 thus go into back of Entity #1

**Rank 4 (0.8)**: Entity #1 brakes, can't swerve to my left due to Entity #2 thus crashing into back of Entity #1

**GT at Rank 4**: Entity #1 changes into my lane due to seeing Entity #2 but does not signal and look due to this my car hits Entity #1

**Rank 1 (0.8)**: Entity #1 decides to change lane due to Entity #2, does not check first before coming into my lane, due to this it hits my car
**Rank 2 (0.6)**: Entity #1 might change lane into my lane without checking blindspot. Entity #1 proceeds with lane change, won't stop in time due to speed and Entity #2 and crash into rear of Entity #1.
**Rank 3 (0.3)**: Entity #1 and Entity #2 converge in the same lane, colliding with each other; due to this, my car won't stop in time and hits both Entity #1 and Entity #2.

(a)  (b)

Figure 13: Example of image-to-text retrieval by our best-performing model, including the annotated hazard (GT) and its rank, alongside the other top three candidates. Each candidate rank is indicated as **Rank n** with the parentheses indicating its ChatGPT similarity to the GT.

## D  More Qualitative Analyses

### D.1  Successful Cases

Figure 13 illustrates two examples of successful cases. In the image shown in Fig. 13(a), the ground-truth explanation is ranked 1st, indicating a correct retrieval. In the case of Fig. 13(b), although the ground-truth explanation is ranked 4th, the top-ranked example has a similar meaning. Therefore, this can still be considered a successful retrieval. Our ChatGPT similarity score, provided within parentheses next to Rank $n$, effectively captures this similarity.

### D.2  Challenging Cases

Figure 14 shows challenging examples of image-to-text retrieval. In the scene shown in Fig. 14(a), the ground-truth explanation expects our car to turn left and collide with Entity #1. The explanations ranked 1st and 2nd propose that our car continues straight while Entity #1 makes a left turn, resulting in a collision. Both explanations present valid hazard hypotheses. Figure 14(b) showcases an example where the ground-truth explanation is ranked very low, specifically at the 277th position. This discrepancy might be due to the term 'red lights' being used to refer to a specific concept, the reverse ramp of a bus, rather than its typical meaning of a traffic signal. Nonetheless, the explanations ranked 1st, 2nd, and 3rd convey meanings that are highly similar to the ground-truth explanation: the self-car cannot stop and collides with the bus.

Figure 15 highlights additional types of challenges. In the scene image shown in Fig. 15(a), our model assigned a very low rank to the ground-truth explanation. We discovered that our model tends to favor shorter hazard explanations and struggles to accurately rank longer and more complex sentences.

Furthermore, we encountered different types of challenges that may arise from using a single image instead of a video or multiple frames. In Fig. 15(b), the car in front of ours appears to be turning left towards our left side. However, our model seems to assume that the car is moving backwards

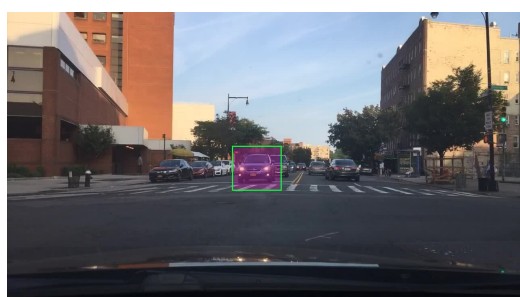 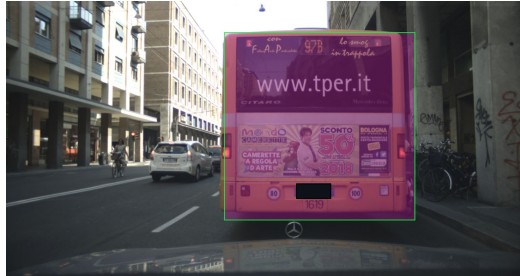

**GT at Rank 596**: My car will destroy Entity #1 while turning left and blocking our path unexpectedly.

**GT at Rank 277**: Entity #1 would stop showing red lights at the back. Due to speed, we would hit Entity #1

**Rank 1 (0.8)**: Entity #1 makes left turn without signaling, misjudges my speed going forward and we collide at intersection

**Rank 2 (0.4)**: Entity #1 decides to make a left turn instead but does not signal, due to this my car won't stop in time and hits Entity #1

**Rank 3 (0.6)**: Entity #1 decides to make turn, but does not factor my speed going forward, Entity #1 my car collide at intersection

**Rank 1 (0.8)**: For the given low speed level, we're following Entity #1 at a close range. We would cause our car to hit Entity #1 in the back as it would apply the brakes unexpectedly.

**Rank 2 (0.8)**: Due to heavy traffic, Entity #1 would certainly stop; following too closely, we would not be able to stop in time.

**Rank 3 (0.8)**: Following Entity #1 back to back, this could make us hit Entity #1 as Entity #1 brakes at a close distance to make a reliable stop.

(a)            (b)

Figure 14: Examples of **challenging cases** of image-to-text retrieval by our best-performing model, including the annotated hazard (GT) and its rank, alongside the other top three candidates. Each candidate rank is indicated as **Rank n** with the parentheses indicating its ChatGPT similarity to the GT.

149 and turning right in front of our car. Consequently, it retrieved hazard explanations that predict the
150 convergence of both cars into the same lane, resulting in a collision.

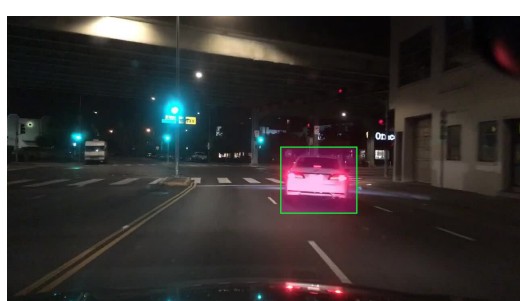

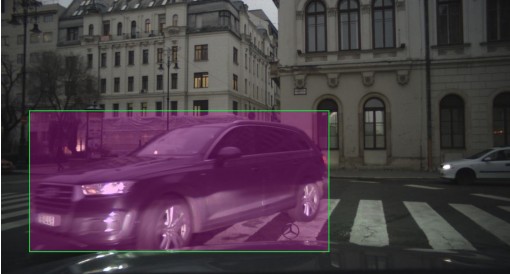

**GT at Rank 340**: I'm turning right in a wide swing. Entity #1 is put on brakes to turn right across the road. I am at full speed, so due to the sudden brake of Entity #1, I will clip the back side of Entity #1.

**Rank 1 (0.4)**: Entity #1 decides to change lane, does not see my car on right lane and factor my speed, due to this Entity #1 collides with my car
**Rank 2 (0.4)**: Entity #1 changes lane to my right, does not signal due to this it hits my car

**Rank 3 (0.7)**: Entity #1 changes lane without signaling and checking mirrors and speed, due to this, i will not stop in time and hits  Entity #1

**GT at Rank 131**: Entity #1 is clipping our car as they misjudge the distance while turning

**Rank 1 (0.2)**: as my car speed is 15 km/h. when my left side Entity #1 was suddenly turn my way, my car will demolish the Entity #1

**Rank 2 (0.6)**: Entity #1 and my car going forward, we both converge into same lane ahead thus colliding with each other

**Rank 3 (0.3)**: Entity #1 is coming right into us giving us no time at 15km/h to stop

(a)                                    (b)

Figure 15: Examples of **challenging cases** of image-to-text retrieval by our best-performing model, including the annotated hazard (GT) and its rank, alongside the other top three candidates. Each candidate rank is indicated as **Rank n** with the brackets containing its ChatGPT similarity to the GT.

## E   Details of Experimental Setup

### E.1   Architecture of CLIP-Based Baselines

The CLIP's extended baselines explained in the main paper all employ the grounded encoders, which consist of two standard transformer layers. These layers sequentially arrange self-attention and cross-attention sub-layers with 512 dimensions divided into 8 attention heads and a dropout rate of 0.1. To enhance positional awareness, we employ relative position embeddings [6], with a maximum distance of 128 and a total of 32 buckets.

### E.2   Loss Functions

For the contrastive loss, we follow the training method of the original CLIP [5]. Specifically, we create mini-batches, each of which consists of a certain number of image-text pairs, $(\tilde{x}_i, h_i)$ $(i = 1, 2, \ldots)$. We have positive pairs $(\tilde{x}_i, h_i)$ and negative pairs $(\tilde{x}_i, h_j)$ with $i \neq j$ within each mini-batch, maximizing the cosine similarity $s$ for the positive pairs while minimizing $s$ for the negative pairs.

For the image-text matching loss, we randomly sample a mismatched pair in half of the image-text pairs created during training. The ITM head maps the corresponding class token into a binary logit for computing binary cross-entropy loss. Only the class token of text features is passed into an Image Text Matching (ITM) head to enable learning the match between the input image-text pair.

If a matching loss is present, the overall loss is the sum of the contrastive loss and the matching loss; otherwise, only the contrastive loss is considered.

We follow [2, 3] to finetune UNITER in image-text retrieval mode by maximizing the margin between the cosine similarity scores between the positive image-text pairs. For fine-tuning BLIP on our image-retrieval tasks, we adopt a similar procedure as used for fine-tuning our CLIP baselines.

### E.3   Training Methods

In our training process, all the models are initialized with its corresponding pretrained weights and finetuned over 15 epochs. We employ a learning rate of $10^{-5}$ and utilize the AdamW optimizer [4] with a linear warmup scheduler in the first 1,000 iterations. Additionally, we utilize a technique called exponential moving average (EMA) with a decay rate of 0.9999 to train all our models, aiming to smoothen the training process and improve the stability of the models. We use an early stopping criterion to determine the optimal stopping point for fine-tuning.

With the exception of models utilizing ViT-L/14, the images undergo resized to a square dimension of $224 + 16$ before being randomly cropped to a size of $224 \times 224$. Conversely, for baseline models that incorporate ViT-L/14, the images are resized to a square dimension of $336 + 16$ and subsequently randomly cropped to a size of $336 \times 336$. We apply the color jitter augmentation with a brightness value of 0.5, hue value of 0.3, and saturation value of 0.3 before highlighting the regions of interested entities in the image. It is noted that we exclude the horizontal flip augmentation to maintain spatial consistency.

## F  Ablation Test with Image/Text Inputs

As stated in the main paper, we can employ the DHPR dataset to create various tasks with different levels of difficulty. Building upon the image-to-text/text-to-image retrieval framework, we conducted additional experiments to examine the influence of input formats. In particular, we varied the format of each of a scene image and a hazard explanation to assess their effects on retrieval performance. In all the experiments, we used the extended CLIP baseline with dual grounded encoders and trained it on samples with new input formats.

**Image Input**    To design different formats for image inputs, we extended the approach proposed by Hessel et al. [3], Specifically, we introduced four types of modified inputs: position only,' no entity,' no context,' and only context,' as illustrated in Fig. 16. Subsequently, we trained and tested the model using each of the new datasets with different formats.

The results are presented in Table 4. Notably, when only the positions of entities were shown ('position only'), a significant decline in performance occurred, with the average rank dropping to over 200. This outcome is reasonable as the model lacks visibility of the visual entities or context; our model utilizes this visual information properly to make inferences. Furthermore, excluding any direct specification of entities ('no entity') also led to considerably poorer performance. This result highlights the necessity for the models to accurately identify the entities present in the given hazard explanations to accurately estimate the similarity between an input image and an explanation.

Interestingly, when the context was removed from the input images ('no context'), relatively better results were obtained, with retrieval ranks ranging from 86.8 to 93.0. However, these ranks were slightly but noticeably worse than the baseline. Conversely, employing only the context information ('only context') yielded significantly worse results compared to removing context alone. These outcomes emphasize the importance of both the visual entities and their surrounding context in making accurate predictions.

**Text inputs**    In terms of text input formats, we add the descriptions of visual entities to hazard explanations. It's important to note that these descriptions have not been utilized in our previous experiments, although DHPR includes them as part of its annotations. Specifically, we enhance the hazard explanation for each sample by incorporating the descriptions of all visual entities into the corresponding section of the explanation. This involves replacing the first occurrence of "Entity #$n$" with "Entity #$n$, <its description>." For example, the following explanation

"Entity #1 decides to go behind of Entity #2 to cross street misjudges my speed, can't stop in time and hits Entity #1"

changes into

"Entity #1, cyclist on right side by sidewalk, decides to go behind of Entity #2, white car in front of my car, to cross street misjudges my speed, can't stop in time and hits Entity #1."

We employed two experiments for training and testing the model. The first experiment involved training the model using the comprehensive format of explanations mentioned above and testing it on the original format of explanations (without the descriptions). This experiment aimed to improve the model's performance on the test split while maintaining the same experimental setting as before.

The second experiment involved testing the same model on the comprehensive explanation format, which includes the additional descriptions. This experiment aimed to evaluate the impact of incorporating these descriptions. However, it is important to note that obtaining the descriptions requires inference and is not freely available. Therefore, we consider this case as an "oracle" scenario.

The lower block of Table 4 presents the results. When evaluating the model trained on comprehensive explanations on the original explanations without the entities' descriptions, it performs significantly worse than the baseline. This decline in performance can be attributed to the disparity in the format

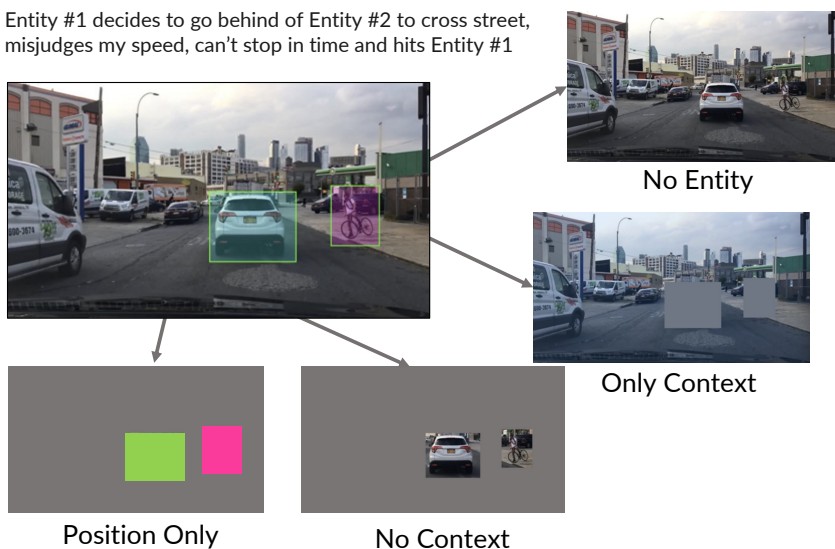

Figure 16: Illustrations of our image input ablations, which are conducted by drawing in pixel-space directly, following [3].

Table 4: Ablation results with varying input data, in which we trained the same baseline with dual encoders in all cases.

| Model | Text-to-Image | | Image-to-Text | |
|---|---|---|---|---|
| | **Direct** | **Indirect** | **Direct** | **Indirect** |
| Baseline | 74.8 | 70.2 | 69.2 | 64.3 |
| Image: position only | 268.2 | 272.1 | 261.4 | 282.3 |
| Image: no entities | 184.4 | 173.3 | 168.3 | 187.2 |
| Image: no context | 86.8 | 96.7 | 88.6 | 93.0 |
| Image: only context | 121.9 | 102.4 | 126.7 | 108.7 |
| Comprehensive text (train only) | 137.5 | 144.2 | 146.7 | 131.3 |
| Comprehensive text (train & test) / **Oracle** | 22.2 | 21.7 | 24.7 | 22.1 |

between the training and test data, indicating that the intended aim was not achieved. However, when the same model is evaluated on the test data using the comprehensive format, a significant improvement is observed, with ranks averaging around 20. We attribute this improvement mainly to the model's ability to associate the added entities' descriptions with the contents of the images. It is why we termed the setting 'oracle.'

# G Evaluating Similarity of Hazard Explanations Using ChatGPT

We utilize ChatGPT (gpt-3.5-turbo) to evaluate the similarity of different hazard explanations, which may include a ground truth explantion, for each scene image. The evaluated scores are used to calculate the NDCG score, as explained in the main paper. We used the following query messages and We received the assistant answer in the "index: relevancy score" format.

> **Messages**
> messages = [
> "role": "system", "content": system_instruction ,
> "role": "user", "content":content
> ]

We designed the system instruction prompt and user content prompt as follows. Note that the instruction specifies several criteria for adjusting a score, which must be in the range of 0 and 1. Additionally, employing a few-shot in-context learning, we provide a single question-and-answer example.

> **System Instruction Prompt**
> As an AI assistant, I will provide similarity scores between sentences based on the criteria you mentioned. The scores will range from 0 to 1, where 0 indicates no similarity and 1 indicates identical meaning. The similarity score will be adjusted based on the following criteria, in order:
>
> If "Entity #$n$" (referring to different objects) does not exist in the reference sentence, the similarity score will be reduced.
> If the scenario is not similar, the similarity score will be reduced.
> If the relative position of the accident is not similar, the similarity score will be reduced.
> If the accident reason is not similar, the similarity score will be reduced.
> If the verb is not similar, the similarity score will be reduced.
>
> Please note that I will only provide the similarity scores without any explanations.
> Please give the answer in the following format. Name: Similarity Score
>
> ### Question:
> Reference: Entity #1 brakes, i'm following to closely and hits Entity #1
> 0: Entity #1 stops while changing lane, my car is too close and hits Entity #1
> 1: Entity #1 applies brake suddenly and not expecting it, due to this my car hits Entity #1
> 2: Entity #1 applies brakes unreasonably, due to this my car can't stop in time and hits Entity #1
> 3: Entity #2 brakes, i'm following to closely and hits Entity #2
> 4: Entity #1 brakes, following to closely and hits Entity #1
>
> ### Answers:
> 0: 0.6
> 1: 0.8
> 2: 0.7
> 3: 0.0
> 4: 1.0

**User Content**

### Question:
Reference: Entity #1 brakes, i'm following to closely and hits Entity #1
1: Entity #1 cuts into my lane and brakes, due to short stopping distance my car hits Entity #1
2: Entity #2 might get into the lane which will make Entity #1 to apply brakes suddenly which might lead us to hit Entity #1 at this speed.
3: I'm driving my car at a moderate speed. At this time, suddenly, Entity #2 is reducing its speed because of Entity #1. I could not stop my car at this speed; my car will hit Entity #2.
4: Entity #1 comes into my lane; I veer to the right to avoid it, thus hitting Entity #2.
5: Entity #1 brakes, due to speed and close proximity, my car will hit Entity #1
### Answers:

We obtained the following answer for the above query:

**Assistant Answer**
1: 0.5
2: 0.2
3: 0.1
4: 0.0
5: 0.9

## H  License

The image assets from the BDD100K dataset are distributed under the BSD 3-Clause License, while the ECP dataset is governed by the eurocity persons dataset research use license. Our usage of both datasets complies with their respective licenses, and we employ anonymization techniques, such as blurring identifiable faces and license plates, to adhere to regulations governing personal data processing.

The DHPR dataset created in this study is licensed under the Creative Commons Attribution-NonCommercial 4.0 International (CC BY-NC 4.0) license. This license allows others to use, adapt, and distribute the dataset, provided they give appropriate credit to the original creator and do not use the dataset for commercial purposes.