# OpenReview forum: "Visual Abductive Reasoning Meets Driving Hazard Prediction: Problem Formulation and Dataset"
_NeurIPS.cc/2023/Track/Datasets_and_Benchmarks — Submitted to NeurIPS 2023 Datasets and Benchmarks_

### Official Review · Reviewer_24dj · 2023-07-21

**Rating:** 5
**Confidence:** 5
**Correctness:** Check the Limitations for details.
**Clarity:** Check the Limitations for details.

**Strengths:**

- The research topic is relatively interesting, different from recently works still considering LLM from the perspective of perception.
- The dataset construction is relatively easy but completed with annotations from different granularity.
- The benchmark construction is detailed with three newly proposed methods.

**Additional Feedback:**

Check the Limitations for details.

**Documentation:**

I have not see the documentation.

**Ethics:**

None.

**Limitations:**

- During the second annotation step, the authors propose to require annotators imaging the vehicle is traveling at a 1.5x velocity and then considering the potential accident, which might not be reasonable, since the imagination cannot change the image content. A driving scene might be dangerous when 1.5x velocity but not in 1x velocity.
- Experiments:
  - CLIP should be one of the most important baselines, which is not presented in Table 3.
  - All variants in Figure 3 can be considered as a special variant of the BLIP model with ITC and ITM only. I do not understand why there is a significant gap between BLIP and the newly proposed methods. It would be better to also fine-tune the baselines on DHPR instead of just the new methods.
  - Refer BLIP2 and its ViT-L/14 variant after fine-tuning to demonstrate the superiority of the proposed methods.
- I can see the great ambition of authors who are trying to enhance the reasoning ability of current vision-language models, but it's a pity to see the authors finally degrade the task from a challenging reasoning task all the way back to a simple matching task even without perception and open-world requirements.

**Opportunities For Improvement:**

I like the topic of this paper, but it's a pity the authors choose to degrade the whole tasks to just a vision-language matching task. Check the Limitations for details.

**Relation To Prior Work:**

The dataset proposes a more completed annotation scheme.

**Summary And Contributions:**

In order to enhance the ability of current vision-language models for visual abductive reasoning and driving hazard prediction, the authors propose a novel image-based dataset called DHPR consisting of 15K dashcam images with annotation including speed, object grounding and potential hazard annotations. For benchmark construction, the authors evaluate two reprehensive baselines and further propose their own models, which demonstrate superior performance.

---

> ### Author Response · Authors · 2023-08-19
>
> **Documentation: I have not see the documentation.**
>
> Thank you for the feedback. We believe some reviewers might have missed our supplementary material due to OpenReview's interface. To access it, go to our submission's full page, look below the abstract for 'Supplementary Material:' with a down arrow icon, and click it for the PDF. Please also see our answer to Reviewer s9mr for a brief summary of our supplementary material.
>
> **Concerns about the second annotation step.**
>
> Please note that our study focuses on potential hazards, which are hypothetical in nature. Our goal is to develop an AI system that can predict these same hypothetical hazards as identified by the annotators.
>
> Given the challenges in acquiring images of traffic scenes leading up to actual accidents (as discussed in the main paper), we use images from accident-free scenes. (It is also noteworthy that not every real-life accident is predictable; a fraction of them are caused by unforeseen events.) Consequently, due to the inherent safety of these scenes, it can sometimes be challenging for annotators to hypothesize potential hazards. To facilitate this process, we ask annotators to assume a 1.5x velocity, which, based on our observations, aids them in identifying potential hazards. We believe that this velocity discrepancy will not pose a significant issue, given that the hazards being annotated are purely hypothetical. While the discrepancy may impact the real-world effectiveness of the resulting models, we believe that the fundamental problem structure remains valid, i.e., geometrically understanding a traffic scene and the relationship between its multiple entities, and predicting potential risks.
>
> **CLIP should be one of the most important baselines, which is not presented in Table 3.**
>
> The baseline method in Table 3 is indeed CLIP, as we state in the main paper that 'We adopt CLIP [30] as our baseline method, following the approach in [15].' Specifically, we fine-tuned both the image and text encoders on the DHPR dataset.
>
> **Concerns about All variants in Fig. 3.**
>
> The models presented in Fig. 3 are extensions of CLIP, not BLIP. These models augment CLIP, which exclusively uses ITC for training, by also incorporating ITM with various auxiliary heads. The performance disparity between BLIP and the CLIP variants may arise from CLIP being pre-trained on a broader and more diverse collection of image-text pairs compared to BLIP. Please refer to Sec. E in the supplementary material.
>
> **Refer BLIP2 and its ViT-L/14 variant.**
>
> In line with your suggestion, we assessed BLIP2 on the DHPR dataset. Results show BLIP2 improves over BLIP but doesn't surpass the CLIP baseline. The best BLIP2 model attains retrieval rankings between 80-90 in Image-to-Text retrieval, while CLIP ranks around 70. We'll incorporate this table to the manuscript.
>
> |Model|Visual Encoder|Text-to-Image - Direct|Text-to-Image - Indirect|Image-to-Text - Direct|Image-to-Text - Indirect|
> |-|-|-|-|-|-|
> |BLIP|ViT-B/16|153.4|172.1|151.9|176.1|
> |||||||
> |BLIP2 (ITC+ITM+LM)| ViT-L/14|98.9|82.5|94.3|81.1|
> |BLIP2 (ITC+LM)| ViT-L/14|94.8|86.0|95.5|80.8|
> |BLIP2 (ITC+ITM)| ViT-L/14|99.1|92.1|97.5|83.7|
> |BLIP2 (ITC)| ViT-L/14|99.7|84.5|99.1|82.4|
> |BLIP2 (LM)| ViT-L/14|408.5|393.3|354.6|403.0|
> |||||||
> |Baseline (Fine-tuned CLIP)|ViT-B/16|77.2|75.3|78.4|73.3|
> |||||||
>
> **Concerns about task degradation.**
>
> We first want to emphasize that this study's primary contribution is the novel introduction of visual reasoning to driving hazard prediction. This approach holds significant potential for advancing driver assistance and autonomous driving. It can, for instance, predict risks further into the future than current standard methodologies. Please note that enhancing the reasoning capacity of existing vision-language models is not our main objective.
>
> On the point of reducing a reasoning task to a simple matching task (i.e., image/text retrieval), we respectfully differ from the reviewer's perspective if they believe this diminishes our study's value. Utilizing text/image retrieval as a representation is a stepping stone towards more sophisticated solutions. The ultimate objective is to autonomously generate a hazard explanation from a given scene image. The decision to use the retrieval-based formulation was taken because visual abductive reasoning is still in its infancy, and the generation of reasoning texts remains a complex challenge. It's worth noting that the seminal study [15], which introduced visual abductive reasoning and the Sherlock dataset, also utilized this method.
>
> Given the swift progress in LLMs and their synergy with visual encoders, we anticipate that text generation will become increasingly accessible. As this unfolds, our dataset will continue to be relevant for studying the creation of reasoning texts. We believe our research provides a strong foundation for incorporating visual abductive reasoning into autonomous driving and driver assistance systems.

---

### Official Review · Reviewer_YbBX · 2023-07-21
**Review for Submission493**

**Rating:** 6
**Confidence:** 4

**Strengths:**

The task definition is clearly and differentiated from existing research.

**Additional Feedback:**

No additional feedback.

**Clarity:**

For Figure 1, a single image from the proposed DHPR dataset depicting a real-world scenario is sufficient to convey the intended message.

**Correctness:**

The claim in the submission is mostly correct. While this paper simplifies the baseline methods for inferring potential risks from static images, it provides a relatively clear definition of the task overall.

**Documentation:**

The authors have provided sufficient documents.

**Ethics:**

No ethical concerns.

**Limitations:**

The reliance on retrieval methods may have scalability and generality issues in addressing visual reasoning problems.

The DHPR dataset's diversity and representativeness of real-world driving scenarios need to be carefully assessed.

The lack of temporal information in static images could limit hazard prediction accuracy compared to video-based approaches.

The focus on single static images may not fully capture the dynamic nature of driving and real-time hazard prediction scenarios.

**Opportunities For Improvement:**

Although the visual reasoning problems proposed in the submission are difficult to solve directly and can only be temporarily addressed by reducing the difficulty and solving them through retrieval methods, a clear pipeline of related works and specific challenges is encouraged.

**Relation To Prior Work:**

The relation to prior methods have been discussed.

**Summary And Contributions:**

The paper proposes a novel problem of identifying the risky driving scenes from a static image. This problem is a specific application of visual abductive reasoning. Furthermore, the paper proposes several baseline methods and introduces a new dataset called DHPR, which is specifically designed for this task.

---

> ### Author Response · Authors · 2023-08-19
>
> Thank you for your comments. We will revise the manuscript based on your suggestions. Below, we address each of the limitations you highlighted and outline the discussions that will be added to the manuscript and the supplementary material.
>
> **The reliance on retrieval methods may have scalability and generality issues in addressing visual reasoning problems.**
>
> We appreciate your feedback. While we have chosen to present the problem as a text/image retrieval, we think this is an intermediate step. The primary goal is to generate a hazard explanation from a given scene image. We chose the retrieval-based formulation since visual abductive reasoning is an emerging field, and creating reasoning texts is notably challenging. Importantly, the seminal study [15], which pioneered visual abductive reasoning and introduced the Sherlock dataset, utilized the same approach.
>
> Given the swift progress in LLMs and their synergy with visual encoders, we anticipate that text generation will become increasingly accessible. As this unfolds, our dataset will continue to be relevant for studying the creation of reasoning texts. We believe our research provides a strong foundation for incorporating visual abductive reasoning into autonomous driving and driver assistance systems.
>
> **The DHPR dataset's diversity and representativeness of real-world driving scenarios need to be carefully assessed.**
>
> We acknowledge that there is considerable room for improvement in our dataset, especially in the aspects suggested. However, we view our dataset as a valuable initial effort in the field. We intend to refine it using feedback from readers and users. Additionally, we hope our study will motivate other researchers, spurring further advancements in the research.
>
> **The lack of temporal information in static images could limit hazard prediction accuracy compared to video-based approaches.**
> **The focus on single static images may not fully capture the dynamic nature of driving and real-time hazard prediction scenarios.**
>
> We recognize the inherent limitation of predicting hazards from a single static image. While certain hazards require temporal information for precise identification, others can be predicted without it. For instance, flagging down a taxi serves as an example. Our study is specifically tailored to identify hazards of this latter type.
>
> Focusing solely on these types of hazards significantly simplifies the problem. This simplification leads to practical research benefits, such as decreased storage needs and more compact model sizes, making the research more manageable.
>
> We wish to emphasize that our current approach is an intermediate step. After completing single-image-based prediction, our subsequent aim is to tackle video-based prediction. Importantly, video-based hazard prediction can leverage single image-based methods. Without first addressing the relatively simpler single-image prediction, it would be difficult to navigate the complexities of video-based prediction.

---

### Official Review · Reviewer_s9mr · 2023-07-22
**Review of Visual Abductive Reasoning Meets Driving Hazard Prediction: Problem Formulation and Dataset**

**Rating:** 7
**Confidence:** 3
**Correctness:** The claims in the submission are corr…
**Clarity:** The paper is well-written and easy to…

**Strengths:**

1. The paper introduces a novel problem formulation and dataset for visual abductive reasoning in driving hazard prediction, which is an understudied area with high potential for driver assistance and autonomous driving systems. The paper will attract more interest in the new problem.


2. The paper creates a new dataset, named DHPR (Driving Hazard Prediction and Reasoning), which consists of 15K dashcam images of street scenes, each annotated with a car speed, a hypothesized hazard description, and visual entities present in the scene. The dataset covers both direct and indirect hazards, which require different levels of complexity and reasoning.


3. The paper proposes a new method for the new problem.





**Additional Feedback:**

The weaknesses above should be carefully addressed.

**Documentation:**

The details of the data collection are demonstrated. The informantion about the evaluation benchmarks is also provided.

**Ethics:**

There are no ethical concerns in the paper.

**Limitations:**

The authors addressed the limitations of their work.

**Opportunities For Improvement:**

Weaknesses:

1. Experiments and analysis are not enough in this paper. The paper only provided the comparison results in Table 3. More experiments and analysis about the dataset or ablation studies are needed.


2. The statistics and analysis of the proposed dataset should be described detailly.


**Relation To Prior Work:**

The paper provided enough details of the comparison between the paper and previous datasets in Table. 1.

**Summary And Contributions:**

The paper presents a new approach and dataset for predicting and reasoning about driving hazards using dashcam images. The approach formulates the problem as visual abductive reasoning, which requires inferring the most plausible explanation of a potential accident based on a single image. The dataset, named DHPR (Driving Hazard Prediction and Reasoning), contains 15K dashcam images of street scenes, each annotated with a car speed, a hypothesized hazard description, and visual entities involved in the hazard. The paper also proposes a benchmark for the task of temporal article grounding, which aims to align the steps of an article with the segments of a video. The paper evaluates several state-of-the-art methods and baselines on this task, and shows that combining weakly supervised pretraining with strongly supervised finetuning leads to the best performance.

---

> ### Author Response · Authors · 2023-08-19
>
> Thank you for your feedback. We believe some reviewers may not have accessed our supplementary material, due to the complicated interface of OpenReview. For clarity, our supplementary material can be accessed from the full page of our submission (rather than from the summary page). Directly below the paper abstract on this page, there's a line labeled 'Supplementary Material:' accompanied by a down arrow icon. Clicking on this will provide access to the  PDF.
>
> In the supplementary material, we present an analysis of our dataset in
> - Sec. C 'Dataset Analysis'
>
> We also provide ablation tests etc. in
> - Sec. D 'More Qualitative Analyses'
> - Sec. F 'Ablation Test with Image/Text Inputs'
>
> The summarized content of these sections is provided below. For more detailed information, please refer to the supplementary material. We believe these sections adequately address the reviewer's concerns. Additionally, to improve dataset analysis, we have created a new summary table displaying more detailed dataset statistics, which is included below. This table will also be added to the supplementary material.
>
> | Data Attributes | Training Set | Validation Set - Direct | Validation Set - Indirect | Test Set - Direct | Test Set - Indirect |
> |------------------------------------------|------------------|------------------------------|--------------------------------|------------------------|--------------------------|
> | **Data Sample** | 10975 | 1000| 1000| 1000| 1000|
> | **Image sourced from:** ||||||
> | ECP dataset | 6010| 456| 502| 480| 472|
> | BDD dataset | 4965| 544| 498| 520| 528|
> | **Our car speed:** ||||||
> | 15 km/h | 4467| 518| 275| 533| 267|
> | 45 km/h | 3859| 351| 439| 338| 430|
> | 75+ km/h | 2649| 131| 286| 109| 303|
> |||||||
> |**Hazard:**||||||
> | Avg. Length Tokens| 30.9| 26.9| 27.6| 27.0| 27.7|
> | **Position words:**||||||
> | Left| 1347 | 67| 209| 70| 193|
> | Right| 1511 | 59| 168| 66| 168|
> | Front| 1749 | 156| 102| 130| 79|
> | Side| 1563 | 73| 162| 80| 157|
> | Back| 934 | 158| 65| 149| 71|
> | **Accident-related words:** ||||||
> | Hit | 6608 | 558| 577| 602| 578|
> | Crash | 1382| 118| 142| 94| 110|
> | Collide | 643| 56| 61| 52| 63|
> | Clip | 121| 25| 75| 19| 73|
> |||||||
> | **Entity Descriptions:** ||||||
> | **Entities:** ||||||
> | Car | 7427| 822| 613| 848| 630|
> | Pedestrian and Bicycles| 2233| 98| 216| 75| 228|
> | **Orientation words:** ||||||
> | Front | 3921| 535| 210| 509| 233|
> | Side | 2448| 87| 390| 81| 436|
> | Right | 2275| 78| 408| 70| 438|
> | Left | 1719| 68| 299| 63| 310|
> | Ahead| 1552| 224| 184| 210| 167|
> | Road | 1380| 62| 106| 63| 105|
> |Brake|1223| 142| 0| 141| 0|
> |Parked|1027| 0| 156| 0| 157|
> |||||||
>
>
> **The summary of the related sections of the supplementary material:**
>
> Section C provides various statistics of DHPR. Specifically, word usage in the hazard explanations is shown in Table 1 of the supplementary material. The types of entities that the self-car is described as colliding with are detailed in Table 2. Additionally, the distribution of verbs used in the explanations can be found in cloud plots depicted in Figs. 11 and 12 of the supplementary material.
>
> In Section D, we show qualitative analyses of the retrieval results from several chosen scenes, encompassing both successful and challenging cases.
>
> Section F shows ablation tests (Table 3 of the supplementary material.) These tests involve ablating the image input in four distinct manners: 'position only', 'no entity', 'no context', and  'only context'. The results indicate the importance of the color coding for entity identification, alongside the value of the visual features of the entities as well as their backgrounds. We also present the results of enhancing text inputs with entity descriptions.
>
> The supplementary material also provides the summary of the dataset (Sec. A), a detailed explanation of the data collection process (Sec. B), more detailed experimental settings (Sec. E), the method for measuring explanations' similarity using ChatGPT (Sec. G) etc.

---

### Decision · Program_Chairs · 2023-09-22

**Decision:**

Reject

**Comment:**

The paper presents a new dataset for predicting and reasoning about driving hazards using dashcam images, named DHPR (Driving Hazard Prediction and Reasoning), contains 15K dashcam images of street scenes, each annotated with a car speed, a hypothesized hazard description, and visual entities involved in the hazard. Experiments and analysis need to be improved. The meta-reviewer agrees with reviewers that more experiments and analysis about the dataset or ablation studies are needed.